# Loss of meiotic double strand breaks triggers recruitment of recombination-independent pro-crossover factors in *C. elegans* spermatogenesis

JoAnne Engebrecht[1,2*], Aashna Calidas[1], Qianyan Li[1,2], Angel Ruiz[1,3], Pranav Padture[1,4], Neeraj Bhavani Aniyan Bhavana[5], Consuelo Barroso[6], Enrique Martinez-Perez[6], Nicola Silva[5*]

**1** Department of Molecular and Cellular Biology, University of California Davis, Davis, California, United States of America, **2** Biochemistry, Molecular, Cellular and Developmental Biology Graduate Group, University of California Davis, Davis, California, United States of America, **3** Molecular Biology Institute, University of California Los Angeles, Los Angeles, California, United States of America, **4** Molecular, Cellular, and Developmental Biology, University of California Santa Barbara, Santa Barbara, California, United States of America, **5** Department of Biology, Faculty of Medicine, Masaryk University, Brno, Czech Republic, **6** MRC Laboratory of Medical Sciences, London, United Kingdom

* jengebrecht@ucdavis.edu (JE); silva@med.muni.cz (NS)

## Abstract

A key event in meiosis is the conversion of a small subset of double strand breaks into interhomolog crossovers. In this study, we demonstrate that *Caenorhabditis elegans* male spermatogenesis has less robust mechanisms than hermaphrodite oogenesis in regulating crossover numbers. This is not a consequence of differences in meiotic prophase timing, sex chromosome genotype, or the presence or absence of germline apoptosis. Using the cyclin-like crossover marker COSA-1, we show that males are less efficient in both converting double strand breaks into crossover designated events and limiting their number, suggesting weakened crossover homeostasis. Surprisingly, we discovered that significant numbers of COSA-1 foci form at the very end of meiotic prophase in the absence of SPO-11 during spermatogenesis. These COSA-1-marked sites are also independent of homologous recombination, and Topoisomerases I and II. We find that the synaptonemal complex, which holds homologs in proximity, differently modulates COSA-1 enrichment to chromosomes in the absence of SPO-11 in males and hermaphrodites. Together, these findings suggest that males have less robust crossover control and that there are previously unrecognized lesions or structures at the end of meiotic prophase in spermatocytes that can accumulate crossover markers.

**Data availability statement:** All relevant data are within the manuscript and its Supporting Information files.

**Funding:** N.S. is funded by the Czech Science Foundation (GA23-04918S). J.E. is funded by National Institutes of Health grants GM103860 and GM103860S1. A. R. was supported by National Institutes of Health R25GM116690. E.M.-P. is funded by Medical Research Council (MRC) grant MC-A652-5PY60. The funders had no role in study design, data collection and analysis, decision to publish, or manuscript preparation.

**Competing interests:** The authors have declared that no competing interests exist.

## Author summary

Formation of healthy gametes depends on the accurate partitioning of genetic material in the daughter cells through meiosis. A hallmark of meiosis is the establishment of crossovers, which arise from physical exchange of DNA between the parental chromosomes during homologous recombination, and are essential for proper chromosome segregation. Recombination is initiated via the induction of physiological DNA damage by the topoisomerase-like SPO-11 enzyme and its auxiliary factors. Abrogating SPO-11 activity prevents crossover formation, resulting in random chromosome segregation and generation of aneuploid gametes. While the underlying mechanisms of crossover formation are conserved between the sexes, several pieces of evidence indicate extensive sexual dimorphism. In our work we describe novel features of *C. elegans* spermatogenesis that reveal significant differences in the regulation of recombination compared to oogenesis. We find that in spermatogenesis crossover-promoting proteins can be recruited to chromosomes even in the absence of SPO-11 activity, a phenomenon not observed in the oogenic germ line. Furthermore, removal of some auxiliary factors important for physiological break formation during oogenesis does not prevent crossover designation in spermatocytes. We show that the synaptonemal complex, tasked with keeping homologous chromosomes in proximity, exerts opposing roles in males and hermaphrodites by promoting and limiting the recruitment of SPO-11-independent crossover factors, respectively.

## Introduction

The meiotic cell cycle consists of a single round of DNA replication followed by two rounds of chromosome segregation to produce haploid gametes for sexual reproduction. In many organisms this is accomplished by the intentional induction of tens to hundreds of lethal DNA double strand breaks (DSBs) by the conserved topoisomerase-like enzyme Spo11 [1,2]. Spo11-induced breaks are subsequently processed predominantly by the homologous recombination (HR) machinery, and a small subset, one to three on each chromosome pair in many organisms, is repaired into interhomolog crossovers (COs) [3,4]. COs provide a physical connection between homologous chromosomes to promote accurate chromosome segregation. Consequently, DSB formation, processing, CO designation, and resolution are essential for meiosis.

CO designation is the process by which a small subset of DSBs is processed into COs. CO designation must be tightly regulated to ensure that each homolog pair receives at least one CO, referred to as CO assurance. At the same time, it is critical to limit their numbers as too many COs can lead to errors in chromosome segregation [5]. CO homeostasis ensures a constant number of COs under conditions where more or fewer DSBs are generated, while CO interference limits formation of closely spaced COs [3,4]. Although the precise mechanisms governing CO regulation are not

understood, the large zipper-like protein structure formed between homologous chromosomes, the synaptonemal complex (SC), has been shown to play a critical role in CO control [6–8].

In metazoans with defined sexes, the products of meiosis —sperm and oocytes— are distinct but each must contribute precisely half of the parental genome to the next generation. Given the necessity for generating haploid gametes in the production of both sperm and oocytes, it is surprising that there is extensive sexual dimorphism in the meiotic program. In organisms with defined sexes where it has been examined, sex-specific differences have been observed in the temporal program of events, the formation, processing and designation of DSBs into COs, the SC, checkpoint signaling, chromosome segregation, and sex chromosome behavior [9–13]. However, little is known about how these events are regulated in a sex-specific manner.

*Caenorhabditis elegans* is an excellent system to elucidate how sex influences the meiotic program. *C. elegans* exist primarily as hermaphrodites – producing sperm in the last larval stage and switching exclusively to oocyte production as adults. Males also exist in the population and undergo spermatogenesis throughout their life span [14]. The genetic basis of sex determination in *C. elegans* is the *X* chromosome to autosome ratio: *XX* worms develop into hermaphrodites, while *X0* worms develop into males [15]. Similar to mammals, during meiosis the single *X* chromosome of males undergoes meiotic sex chromosome inactivation, a process whereby the unpaired sex chromosome accumulates repressive chromatin and is transcriptionally silenced [16].

Like the *C. elegans* hermaphrodite germ line, the *C. elegans* male germ line is arranged such that all stages of meiosis are present and arrayed by stage from the distal to the proximal end of the gonad [17] (Fig 1A). Although fundamentally similar in organization, germline differences exist between the sexes including the time spent in meiotic prophase [18,19]. Additionally, unlike in hermaphrodites, males do not exhibit germline apoptosis either under physiological conditions or in response to errors through checkpoint signaling [20,21].

There is also evidence for differential regulation of meiotic recombination in the sexes [10,22–24]. As there is no direct marker of DSBs in *C. elegans*, immunolabeling of the recombinase RAD-51 in the germ line has been used as a proxy to monitor DSB formation and repair [25]. Interestingly, the pattern of appearance and disappearance of meiotic RAD-51 foci is distinct in the male vs hermaphrodite germ line: RAD-51 appears to load earlier in meiotic prophase, reaches higher steady state levels, and is removed more abruptly in males [10,22,23]. There also appears to be differences in the regulation of CO control: while in oogenesis, there is very strong CO interference and a single CO is detected per chromosome pair in most meioses, chromosome pairs containing two COs have been detected in a few percentages of male meioses, suggesting that males have less robust CO control [24,26,27].

Here we investigate how CO control is regulated in *C. elegans* males. Using cyclin-like COSA-1 as a CO marker [28], we show that the increased number of extra COSA-1 foci in male meiosis does not appear to be a consequence of the time germ cells spend in meiotic prophase, the presence of a single *X* chromosome in male meiosis, or the absence of germline apoptosis in male germ lines. We find that under limiting DSBs, males have less stringent CO homeostasis. Surprisingly, we discovered that COSA-1 and its partners accumulate at the end of meiotic prophase in the absence of DSBs in spermatogenesis. This accumulation is modulated by the SC and suggests that there is a previously unrecognized chromosome structure or lesion at the end of meiotic prophase in spermatocytes where CO factors can bind.

## Results

### COSA-1 numbers are less tightly controlled in *C. elegans* male germ cells

To determine if CO control is regulated differently in hermaphrodites versus males, we monitored the localization of the pro-crossover protein GFP::COSA-1 (from *GFP(glo)::3xFLAG::cosa-1(xoe44)* at the endogenous locus [29], hereafter referred to as *GFP::cosa-1(xoe44)*) [22,28], in hermaphrodites undergoing oogenesis and in male spermatogenesis (Fig 1A). In hermaphrodites, the number of GFP::COSA-1 is almost exclusively six, suggesting that there is a single COSA-1-marked event on each homologous chromosome pair (Fig 1B; [28]). While male germ cells have on average five GFP::COSA-1 foci,

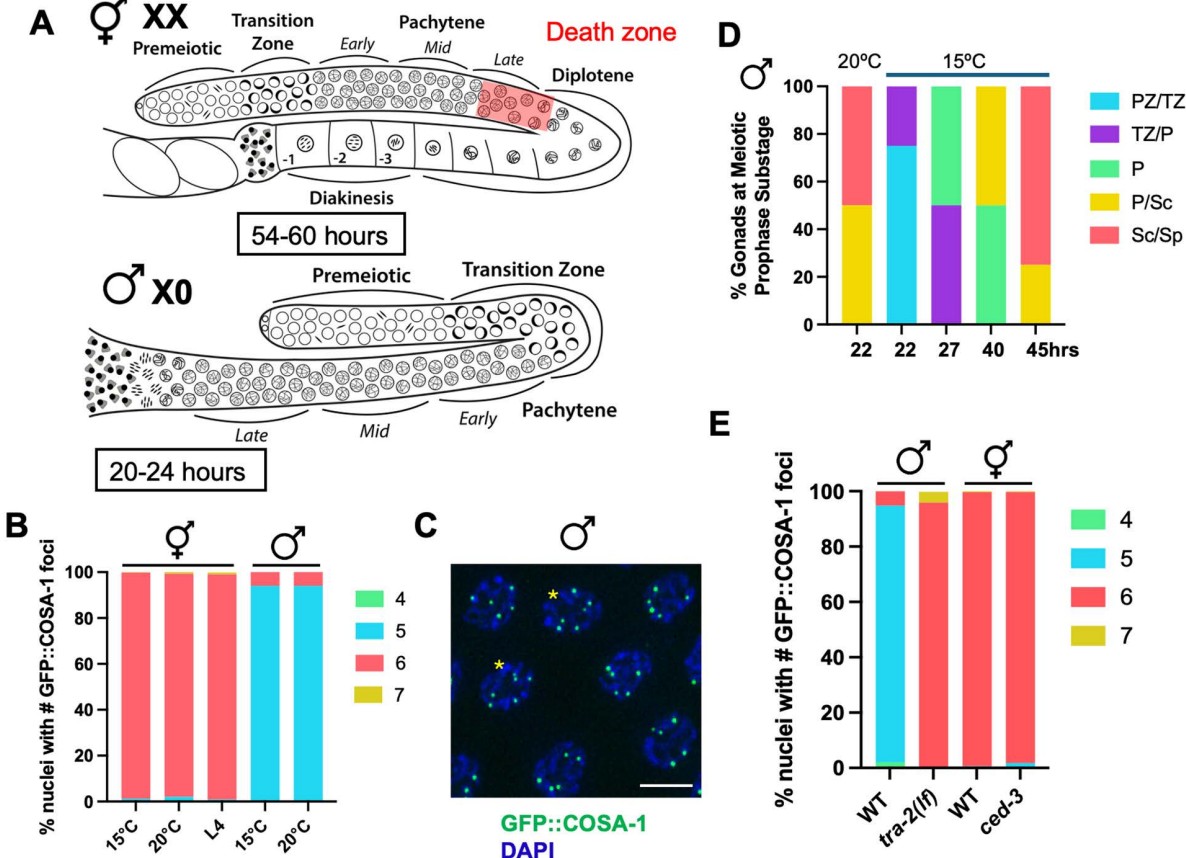

**Fig 1. Sex-specific germline features do not alter GFP::COSA-1-marked events.** (A) Cartoon of the hermaphrodite (*XX*) and male (*X0*) germ lines showing sexually dimorphic features including timing through meiosis (hours in box), *X* chromosome genotype (*XX* vs. *X0*) and the presence of apoptosis in hermaphrodites (red death zone). Different meiotic stages are indicated. (B) Stacked bar graph showing percent nuclei with indicated numbers of GFP::COSA-1 foci in hermaphrodites and males grown at either 15°C or 20°C as well as in L4 hermaphrodites grown at 20°C. Green = 4; cyan = 5; rose = 6; gold = 7. Number of nuclei examined *GFP::cosa*-1 hermaphrodite 15° = 627; 20° = 406; L4 = 321; male 15° = 350; 20° = 371. (C) Male mid-late pachytene nuclei imaged for GFP::COSA-1 fluorescence (green) and counterstained with DAPI (blue). Asterisks denote nuclei with 6 GFP::COSA-1 foci. Scale bar 5μm. (D) Percent gonads at meiotic prophase substage based on S phase labeling at 15° and 20°C. PZ = proliferative zone; TZ = transition zone (leptotene/zygotene); P = pachytene; Sc = spermatocyte; Sp = sperm. PZ/TZ = aqua; TZ/P = purple; P = green; P/Sc = gold; Sc/Sp = rose. (E) Stacked bar graph showing percent nuclei with indicated numbers of GFP::COSA-1 foci in WT and *tra-2(lf)* males and in WT and *ced-3* hermaphrodites. A minimum of 4 germ lines were examined; number of nuclei scored: *GFP::cosa*-1 male = 267, *tra-2(e1095); GFP::cosa-1* male = 254, *GFP::cosa-1* hermaphrodite = = 232, *GFP::cosa-1(xoe44); ced-3(ok2734)* hermaphrodite = 321. Green = 4; cyan = 5; rose = 6; gold = 7.

one for each of the five pairs of autosomes as previously reported [22,24], a few percentage of pachytene nuclei contain six GFP::COSA-1 foci, suggesting that CO numbers are not as tightly regulated in male spermatocytes (Fig 1B and 1C; [10,30]). This pattern of COSA-1 foci correlates well with the number of genetic crossovers observed, where more double COs are detected in male meiosis [24,26,27]. We also examined GFP::COSA-1 in larval stage 4 (L4) hermaphrodites undergoing spermatogenesis and did not observe a class of nuclei containing an extra GFP::COSA-1 focus (Fig 1B). This suggests that COSA-1 foci numbers in male spermatogenesis are less tightly controlled compared to in hermaphrodites.

### Sex-specific germline features do not influence the pattern of COSA-1-marked events

To determine whether sexually dimorphic attributes of the germ line contributed to the differences observed in the number of GFP::COSA-1 foci, we examined GFP::COSA-1 when meiotic prophase timing, *X* chromosome genotype, or germline

apoptosis was perturbed (Fig 1A). We first examined whether the amount of time spent in pachytene affected the number of GFP::COSA-1-marked events as we previously found that meiotic prophase for sperm production in males is approximately twice as fast (20-24h) as it is for oocyte production in hermaphrodites (54-60h) [18]. Further, the biggest difference in timing between the sexes is in pachytene, where CO designation occurs [28,31]. To that end, we incubated male worms at 15°C instead of the standard temperature of 20°C, and followed meiotic progression with S-phase labeling [18,32] as well as monitored GFP::COSA-1. Young adult worms were given a short pulse of 5-Ethynyl-2′-deoxyuridine (EdU) and then incubated at 15°C for various times. Detection of EdU-labeled cells revealed that worms incubated at 15°C underwent meiotic prophase with ~2x slower kinetics than worms incubated at 20°C (15°C = 36-48h; 20°C = 20-24h; Fig 1D). This is close to the time that hermaphrodites spend in meiotic prophase for oocyte production (48-60h) at 20°C [18]; however, the increased time for germ cell nuclei to transit the gonad occurred at both the mitotic region of the germ line (15°C = ~20h vs. 20°C = ~12h) and in pachytene (15°C = ~18h vs. 20°C = ~9h). Analysis of GFP::COSA-1 revealed that the same percentage of nuclei with six COSA-1 foci was observed in male worms incubated at 15° as at 20°C (Fig 1B), suggesting that it is not the speed of meiosis in males that impairs efficient channeling of DSBs into COSA-1-marked events. However, we cannot eliminate the possibility that slowing pachytene to a greater extent would alter how DSB sites are designated into COSA-1-marked events.

Another difference between males and hermaphrodites is the presence of the single, unpaired X chromosome in males vs. the paired XXs in hermaphrodites. Previous work has shown that the X chromosome of males is a substrate for SPO-11-induced DSBs; however, GFP::COSA-1 does not accumulate on the X as no interhomolog COs can form [22,23]. Lack of COs on a subset of chromosomes has been shown to lead to the inter-chromosomal effect, whereby there is an increased likelihood of paired chromosomes receiving additional COs [33–35]. To determine if the presence of the unpaired X chromosome of males promotes additional COSA-1-marked events on other chromosomes, we examined GFP::COSA-1 in strains carrying the tra-2(lf) sex determination mutation that transforms XX animals into males [36]. Previously we found that tra-2 XX males behave similarly to wild-type X0 males with respect to meiotic prophase timing and lack of apoptosis [23]. Like XX hermaphrodites, XX males had on average six GFP::COSA-1 foci, corresponding to the six pairs of homologous chromosomes. However, unlike hermaphrodites but similar to X0 males, XX males had a class of nuclei containing an extra GFP::COSA-1 focus, in this case seven (Fig 1E). Thus, the presence of the unpaired X chromosome in males does not lead to the formation of extra GFP::COSA-1 foci and suggests that some other aspect of male spermatogenesis is permissive for the formation of extra COSA-1-marked events.

In hermaphrodites, physiological germline apoptosis is important for the formation of quality oocytes and apoptosis is also induced in a checkpoint-dependent manner in response to errors to cull defective germ cells; however, neither physiological nor checkpoint-dependent apoptosis occur in the male germ line [20,21,37,38]. We next tested the hypothesis that germline apoptosis in hermaphrodites culls nuclei with greater than six GFP::COSA-1 foci, while in males there is no apoptosis so nuclei with extra foci are not eliminated. To that end, we examined GFP::COSA-1 in ced-3 mutant hermaphrodites. CED-3 is the cell death executor caspase required for cell death; mutation of CED-3 blocks apoptosis [39]. No differences in GFP::COSA-1 foci were observed between wild-type and ced-3 mutant hermaphrodites suggesting that absence of nuclei with extra COSA-1-marked events in hermaphrodites is not due to apoptotic cell death (Fig 1E). Thus, germline intrinsic features do not appear to contribute to the formation of the extra COSA-1-marked events observed in males.

## Conversion of DSBs into COSA-1-marked events is not as tightly regulated in male spermatocytes vs. hermaphrodite oocytes

As neither timing, X chromosome genotype, nor apoptosis appear to affect GFP::COSA-1-marked events, we examined whether it is the nature of the conversion of DSBs into COs that is different in hermaphrodites and males. Previous studies have shown that the kinetics of the recombinase RAD-51 are different in hermaphrodite versus male germ cells, with male

germ lines displaying earlier appearance and a narrower peak of RAD-51 foci in meiotic prophase than hermaphrodite germ lines [10,22,23]. Further, studies by the Villeneuve lab revealed that in hermaphrodites there are mechanisms to ensure a single COSA-1-marked recombination site on each homolog pair under both limiting and excess DSBs [28]. To examine whether males also both maintain and limit COSA-1-marked recombination events in the presence of different levels of DSBs, we conducted an irradiation (IR) dose-response analysis to investigate the relationship between the number of DSBs and COSA-1-marked sites. For these experiments, we exposed *spo-11(ok79)* mutant worms, which do not form DSBs and express the same GFP::COSA-1 transgene as in the hermaphrodite studies [*meIs8[unc-119(+) Ppie-1::GFP::cosa-1*] [28], to different IR doses and then assessed GFP::COSA-1 foci in mid to late pachytene nuclei fixed at 8h post-IR. In the absence of IR there were very few GFP::COSA-1 foci except some foci in very late pachytene and condensation zone nuclei (Figs 2A and S1A, 0Gys, bracket, see below), while there were abundant GFP::COSA-1 foci following exposure to 100Gys IR (Figs 2A and S1A, 100Gys). Quantification revealed that in contrast to hermaphrodites where there were 75.4% nuclei with six GFP::COSA-1 foci at 10Gys, which is estimated to correspond to four DSBs/chromosome pair [28], only 31.7% of nuclei have five GFP::COSA-1 foci at this dose in male germ cells (Fig 2B). Graphing the mean number of GFP::COSA-1 per IR dose revealed that hermaphrodites have a significantly higher slope than males between 0–10Gys (0–4DSBs)

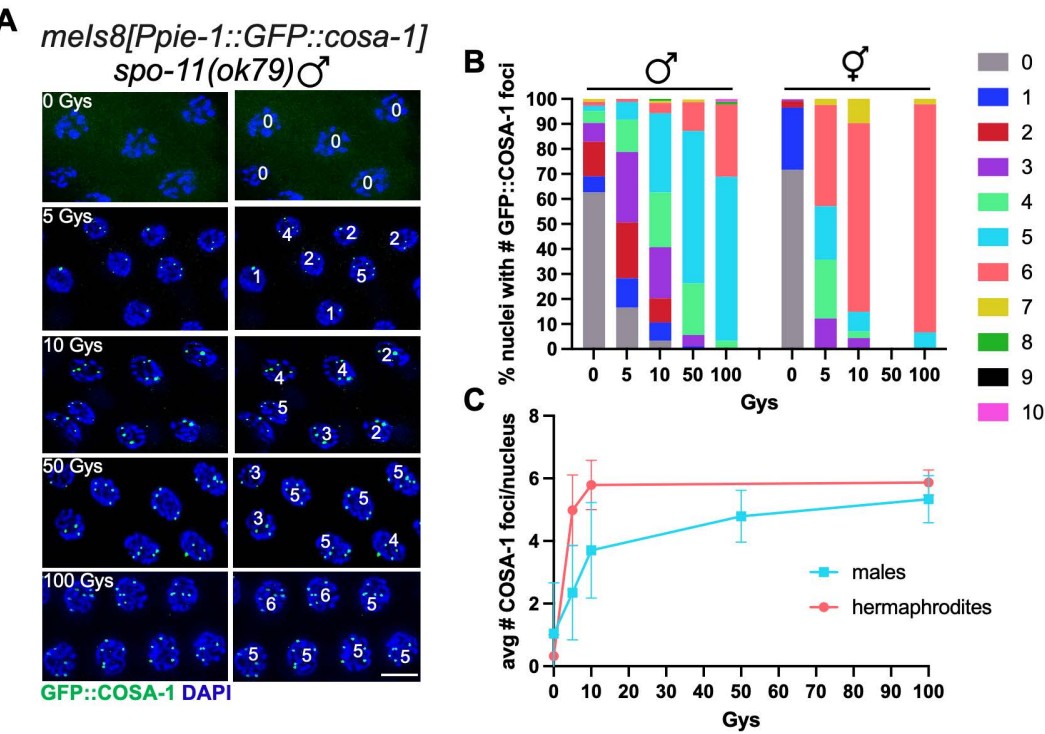

**Fig 2. Conversion of DSBs into GFP::COSA-1 foci is not as tightly regulated in male spermatogenic vs. hermaphrodite oogenic germ cells.** (A) Paired fluorescent images showing GFP::COSA-1 fluorescence (green) counterstained with DAPI (blue) in late-pachytene nuclei from [*meIs8[unc-119(+) pie-1promoter::GFP::cosa-1]; spo-11(ok79)*] germ lines exposed to the indicated Gys of IR, fixed 8h post-IR, with numbers of foci in each nucleus indicated on the right. Scale bar 5 μm. (B) Stacked bar graph showing percent nuclei with indicated numbers of GFP::COSA-1 foci in [*meIs8[unc-119(+) pi-1promoter::GFP::cosa-1]; spo-11(ok79)*] males and hermaphrodites following exposure to indicated Gys of IR. A minimum of three germ lines were examined for each condition except no hermaphrodites were analyzed for 50Gys; the number of nuclei examined are as follows: 0Gys = 187 male, 214 hermaphrodite; 5Gys = 85 male, 98 hermaphrodite; 10Gys = 123 male, 136 hermaphrodite; 50Gys = 87 male, 0 hermaphrodite; 100Gys = 90 male, 92 hermaphrodite. Number of GFP::COSA-1 foci: Grey = 0; blue = 1; red = 2; purple = 3; neon green = 4; cyan = 5; rose = 6; gold = 7; green = 8; black = 9; magenta = 10. (C) Graph showing relationship between IR dose and the mean number and standard deviation of COSA-1 foci per nucleus. Slope between 0-10Gys: hermaphrodites: 0.55, males: 0.26; p < 0.0001 using linear regression.

(hermaphrodites: 0.55 vs. males: 0.26; p<0.0001; Fig 2C), suggesting that hermaphrodites are more efficient at converting limiting IR-induced breaks into COSA-1marked events. There was also a significantly higher number of nuclei with an extra GFP::COSA-1 focus in males compared to hermaphrodites at 100Gys (28.9% six foci in males vs. 2.2% seven foci in hermaphrodites; Fig 2B) suggesting that males have a higher capacity to convert excess DSBs into an additional COSA-1-marked event. Together, these results suggest that CO homeostasis is less efficient in the male germ line.

**Removal of SPO-11 causes the appearance of COSA-1 foci in late pachytene/condensation zone spermatocytes**

We next investigated the temporal dynamics of SPO-11-mediated DSBs in males by analyzing the loading of the recombinase RAD-51 following depletion of SPO-11, taking advantage of the *spo-11::AID::3xFLAG* allele and the spatiotemporal organization of the germ line. The *spo-11::AID::3xFLAG* allele (referred as *spo-11::AID* hereafter) is functional and SPO-11::AID is efficiently targeted for destruction in the presence of auxin and *SUN-1* promoter driven *TIR1* (*ieSi38* [*psun-1::TIR1::mRuby::sun-1 3' UTR*]) [32,40]. We exposed *spo-11::AID* worms to auxin for different times, dissected, immunolabeled for RAD-51, and quantified RAD-51 foci by dividing the germ line in six equal regions spanning the transition zone to entry into the condensation stage (Fig 3A). One hour exposure to auxin sufficed to elicit disappearance of ~40–60% of the RAD-51 foci throughout the transition zone and in early-pachytene stage nuclei in the *spo-11::AID* male worms (Fig 3A, zones 1 and 2), where the majority of RAD-51 are observed. Additional exposure to auxin for 2 or 4h further reduced RAD-51 foci formation throughout the gonad. We also performed exposure to auxin for 24h, which abrogated essentially all RAD-51, suggesting efficient depletion of SPO-11 in the male germ line (Fig 3A).

In hermaphrodites, SPO-11::AID depletion in late meiotic prophase blocks CO formation as analyzed by the endogenously tagged and functional OLLAS::COSA-1 as well as bivalent (chiasmata) formation at diakinesis [32]. We next examined COSA-1 localization following 24h on auxin, which based on RAD-51 immunostaining completely depleted SPO-11 in the male germ line (Fig 3A and 3B). Following auxin depletion for 24h, OLLAS::COSA-1 foci were absent from germ cells except in very late pachytene nuclei and extending into the condensation zone, where a very broad distribution of OLLAS::COSA-1 foci was observed (0–10 foci; Fig 3B). To ensure that this was not a consequence of incomplete auxin depletion or due to the nature of the *cosa-1* allele, as different tagged alleles have different detection sensitivities [30], we examined *spo-11(ok79)* mutant males expressing *meIs8[unc-119(+) Ppie-1::GFP::cosa-1]* or *GFP::cosa-1(xoe44)* from the endogenous locus and also observed nuclei with several GFP::COSA-1 foci in late pachytene/condensation zone nuclei (S1A-S1D Fig). To determine whether the accumulation of GFP::COSA-1 was unique to males, or a feature of spermatogenesis, we examined L4 *GFP::cosa-1(xoe44); spo-11(ok79)* hermaphrodite germ lines, which are undergoing spermatogenesis. As with males, we observed GFP::COSA-1 in late pachytene/condensation zone in the absence of SPO-11 (S1B and S1C Fig). Thus, the broad distribution of COSA-1 foci at the end of meiotic prophase in the absence of SPO-11 is observed regardless of how SPO-11 is removed and which tagged COSA-1 allele is used for detection and is specific to spermatogenesis in both males and L4 hermaphrodites.

To determine if other CO markers are also enriched with COSA-1 in the absence of SPO-11, we examined the cyclin-dependent kinase 2, CDK-2::HA, which forms a complex with the cyclin-like COSA-1 [31]. As at *bona fide* CO sites, CDK-2::HA co-localized with COSA-1 to foci in late pachytene/condensation zone nuclei in the absence of SPO-11 (Fig 4A). We also examined a member of the BTR (Bloom, Topoisomerase IIIa, RMI proteins) complex, the RMI ortholog, RMH-1, which has been shown to localize to six foci along with COSA-1 during oogenesis [41]. As in hermaphrodites, RMH-1 forms several foci in early-mid pachytene cells (presumably labeling most recombination intermediates) and is then recruited at the presumptive CO sites in late pachytene nuclei in males undergoing normal meiosis (S2 Fig). Similar to what we observed for COSA-1, depletion of SPO-11 resulted in the formation of GFP::RMH-1 foci in late pachytene nuclei as well, which extensively co-localized with COSA-1 (Fig 4B). Together, these data suggest that COSA-1 and its partners accumulate at chromosomal sites in late spermatocyte pachytene/condensation zone nuclei in the absence of SPO-11.

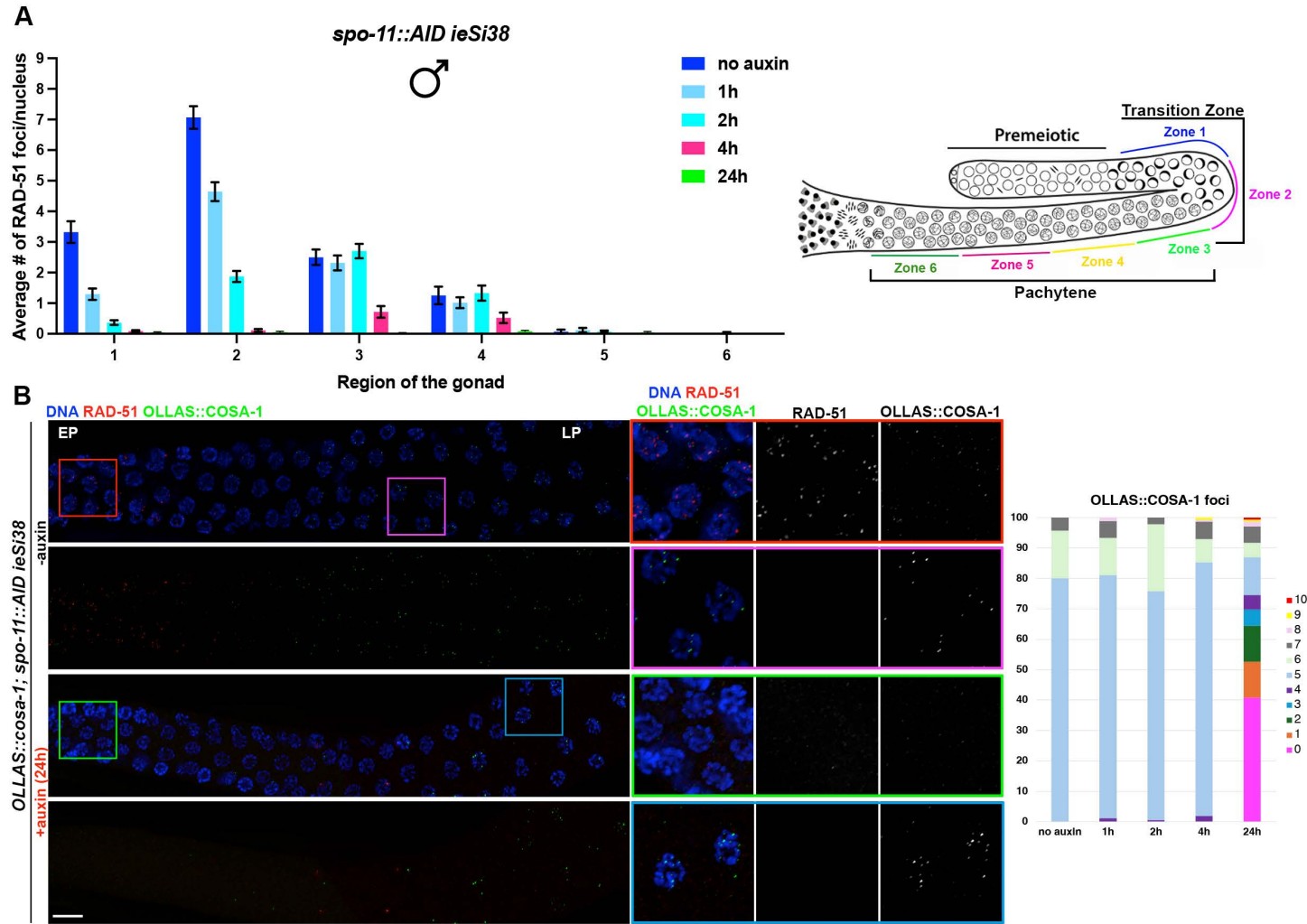

**Fig 3. Pro-CO factors form foci independently of SPO-11 in the male germ line.** (A) Left: quantification of RAD-51 foci number throughout the male germ line at the indicated times of exposure to auxin. At least three gonads were used for quantification. Right: schematic representation of a male germ line depicting the zoning adopted for quantification of RAD-51 foci. The number of nuclei analyzed in untreated controls and after 1h, 2h, 4h, and 24h exposure to auxin is the following: zone 1 (114-119-170-99-152), zone 2 (75-59-87-45-117), zone 3 (56-57-64-46-89), zone 4 (43-65-76-50-83), zone 5 (43-50-57-54-70), zone 6 (38-48-56-42-52). Bars show mean with S.E.M. (B) Left: representative images of male germ lines of the indicated genetic background, before and after exposure to auxin for 24h, immunolabeled for RAD-51 (red), OLLAS::COSA-1 (green) and counterstained with DAPI (blue). EP = Early Pachytene, LP = Late Pachytene. Scale bar 20 μm. Right: quantification of COSA-1 foci at the indicated exposure times to auxin.

## SPO-11-independent COSA-1 foci form in the absence of RAD-51

We next addressed whether the COSA-1 foci observed in the absence of SPO-11 are dependent on HR. We first immuno-labeled RAD-51 in fixed *OLLAS::cosa-1; spo-11::AID* male germ lines from worms exposed to auxin for 24h and examined localization of RAD-51 and OLLAS::COSA-1. We observed very few RAD-51 foci upon *spo-11* depletion (Fig 3B), including in the region where OLLAS::COSA-1 foci are observed, suggesting that chromatin recruitment of COSA-1 occurs independently of prior RAD-51 foci formation. To examine this directly, we generated a new null allele of *rad-51* using CRISPR [42] in the *OLLAS::cosa-1; spo-11::AID* background and examined OLLAS::COSA-1 localization. We first assessed whether our newly generated *rad-51(xoe53)* allele behaved as a predicted *rad-51* loss-of-function allele by analyzing

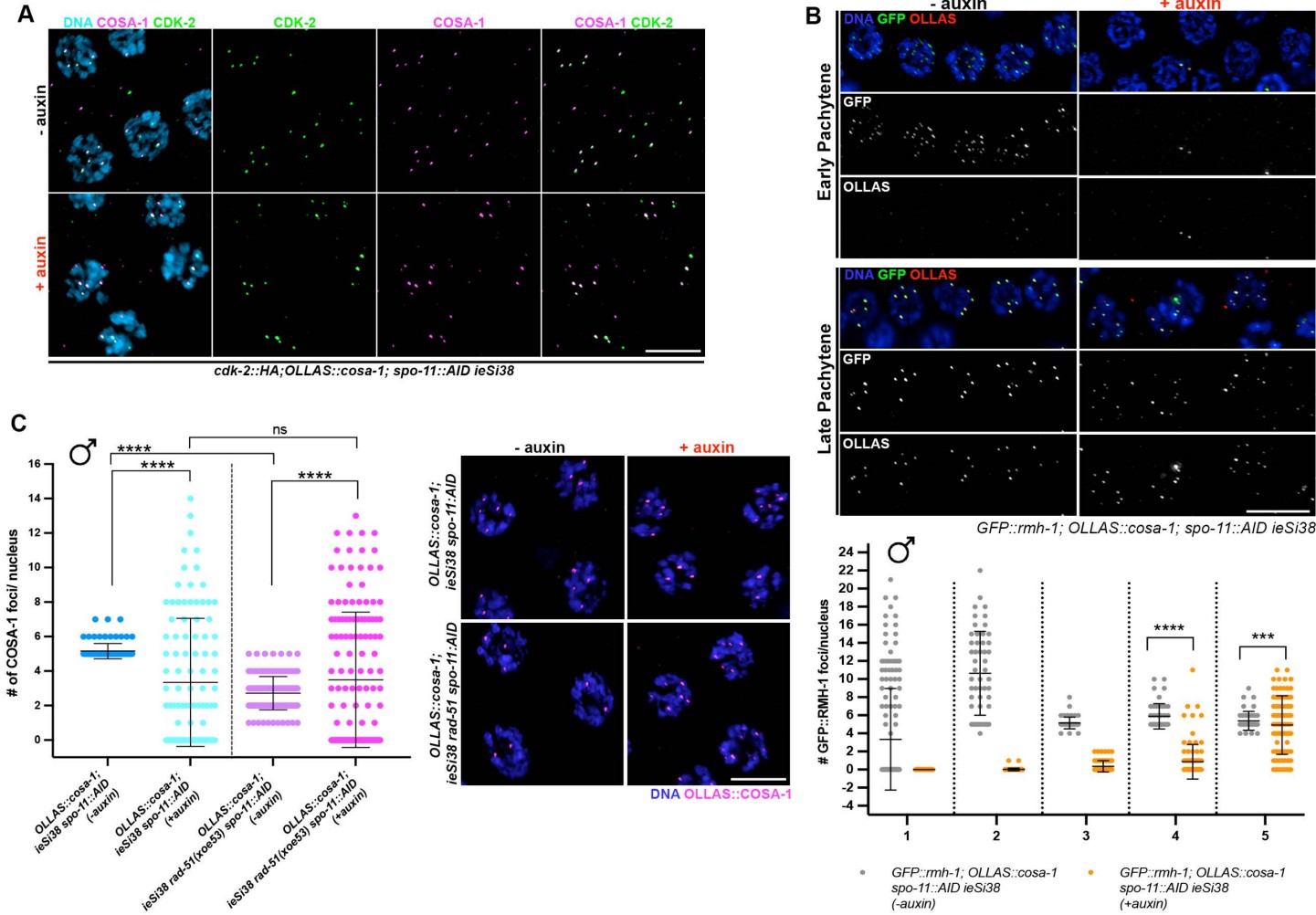

**Fig 4. Recruitment of SPO-11-independent pro-CO factors does not require RAD-51.** (A) Representative images of late pachytene nuclei of the indicated genotype before and after exposure to auxin for 24h, immunoassayed for CDK-2::HA (green) and OLLAS::COSA-1 (magenta) counterstained with DAPI (cyan). Scale bar 5 μm. (B) Top: representative images of GFP::RMH-1 (green) and OLLAS::COSA-1 (red) counterstained with DAPI (blue) in the indicated genotypes, stage and exposure conditions to auxin for 24h. Scale bar 10 μm; Bottom: quantification of GFP::RMH-1 foci number across the male germ line. Gonads were divided into 5 equal regions spanning the transition zone to late pachytene. The number of nuclei analyzed before and after exposure to auxin are respectively: zone 1 (158 - 122), zone 2 (52 - 94), zone 3 (49 - 77), zone 4 (46 – 94), zone 5 (43 – 85). Bars indicate mean with S.D. and asterisks denote statistical significance assessed by Kolmogorov-Smirnov test (****$p < 0.0001$). (C) Left: quantification of OLLAS::COSA-1 foci in the indicated genetic backgrounds and auxin exposure conditions. Bars show S.D. and asterisks denote statistical significance assessed by Kolmogorov-Smirnov test that compares cumulative distributions. ****$p < 0.0001$, ns = *not significant*. Right: representative images of late pachytene cells immunolabeled for OLLAS::COSA-1 (magenta) and counterstained by DAPI (blue). Scale bar 5 μm.

RAD-51 localization as well as diakinesis nuclei, in which impaired HR due to loss of RAD-51 should result in the formation of unstructured/clumped DAPI bodies in hermaphrodites. Immunofluorescence analysis showed no detectable RAD-51 foci in the germ line of hermaphrodites and consistently, analysis of diakinesis nuclei by DAPI staining revealed the presence of unstructured chromatin masses as expected (S3A Fig). Similarly, RAD-51 foci were also absent in the male germ line (S3B Fig). We then proceeded to monitor OLLAS::COSA-1 foci formation in *spo-11::AID rad-51(xoe53)* before and after auxin exposure. Strikingly, while abrogation of RAD-51 function largely impaired COSA-1 loading in hermaphrodites (S3C Fig), it did not alter the number of COSA-1 foci observed in late pachytene/condensation zone nuclei in the

male gonad (Fig 4C), suggesting that the recruitment of COSA-1 to these SPO-11-independent sites is not dependent on HR-mediated processing.

To confirm that the COSA-1 foci observed in the absence of SPO-11 arise independently of HR and therefore do not form *bona fide* COs, we examined the viability of progeny sired by *spo-11(ok79)* or *cosa-1(tm3298)* mutant males. Note that in male worms, chromosome condensation at the end of meiotic prophase precludes analysis of bivalent formation as in hermaphrodites; progeny viability serves as an indirect readout of successful CO formation. As previously reported [43], 74.06±5.82% of progeny sired by *spo-11* mutant males were inviable, presumably due to CO failure (S3D Fig). We observed a similar level of progeny inviability sired by *cosa-1* males (74.68±3.08%; S3D Fig), consistent with an inability to form COs. Taken together, the lack of dependency on RAD-51 and the relatively high inviability of progeny sired by *spo-11* males suggest that the COSA-1 foci recruited in the absence of SPO-11 do not represent interhomolog recombination events.

### Loss of SPO-11 auxiliary proteins does not fully abrogate COSA-1 loading in males

SPO-11 acts in conjunction with several pro-DSB factors across species [44–47], and while SPO-11 itself is highly conserved, these accessory proteins have widely diverged amongst eukaryotes. Besides the MRN/X components, MRE-11 and RAD-50 [48,49], several other factors involved in promoting efficient DSB induction have been identified in *C. elegans*. HIM-17 has been shown to be required for DSB formation presumably by modulating chromatin structure [50]. HIM-5 is required to promote DSB induction specifically on the *X* chromosome [51]. Additionally, DSB-1, -2 and -3 have been identified as distant homologs of yeast Rec114 and Mei4 [52–54], and yeast 2 hybrid experiments have shown that DSB-1 can interact with SPO-11 [52]. Given that all the abovementioned factors are required for DSB induction (although to different extents), we sought to investigate whether removal of these auxiliary proteins would also result in loading of COSA-1 as similarly observed under SPO-11 depletion.

The number of COSA-1 foci was mildly reduced throughout prophase in *him-5(ok1896)* compared to wild-type males. Compared to hermaphrodites [55,56], a significant cohort of nuclei with <5 COSA-1 foci was observed in males (Fig 5A). This phenotype was somewhat surprising and suggests that HIM-5 is important in males to promote DSB formation on the autosomes. Compared to *him-5* mutants, *him-17(ok424)* null worms display a more severe global reduction of SPO-11-dependent recombination intermediates and consistently, we observed about 40% decrease in the average number of COSA-1 foci (Fig 5B). In line with the less pronounced reduction of COSA-1 foci formation, analysis of RAD-51 foci numbers in *him-5(ok1896)* and *him-17(ok424)* males (S4A Fig) also revealed a milder global reduction compared to hermaphrodites [50,51,55].

Analysis of aged *dsb-2* mutant males showed that COSA-1 foci numbers were only mildly reduced compared to wild type (Fig 5C), whereas previously reported [53] aged *dsb-2* mutant hermaphrodites displayed a stark impairment of both RAD-51 and COSA-1 foci (S5A and S5B Fig), indicating sex-specific requirements for DSB-2 activity in promoting recombination. On the other hand, *dsb-1* mutants strongly phenocopied males depleted for SPO-11, as a very wide range of COSA-1 foci was observed, including cells displaying up to 13 COSA-1 foci, in late pachytene/condensation zone (Fig 5D). Compared to the other pro-DSB factors tested, only *dsb-1* mutants recapitulated *spo-11* deficient backgrounds, possibly in line with their putative physical interaction [52]. This suggests that upon reduced physiological DSB levels (as in *him-5*, *him-17* and *dsb-2* mutants), COSA-1 loading can still be largely accomplished during spermatogenesis. However, as observed upon loss of SPO-11, removal of DSB-1 enhances accumulation of COSA-1 in late pachytene/condensation stage in the male germ line.

### SPO-11-independent COSA-1 foci also form independently of Topoisomerases I or II

We next addressed if SPO-11-independent COSA-1 foci are dependent on either topoisomerase 1 or 2. Topoisomerase I (TOP-1) cuts a single DNA strand [57] and accumulates at the end of meiosis in *C. elegans* males [58]. On the other hand, topoisomerase II (TOP-2) cuts both DNA strands and has been shown to have a specific requirement in male meiosis [59].

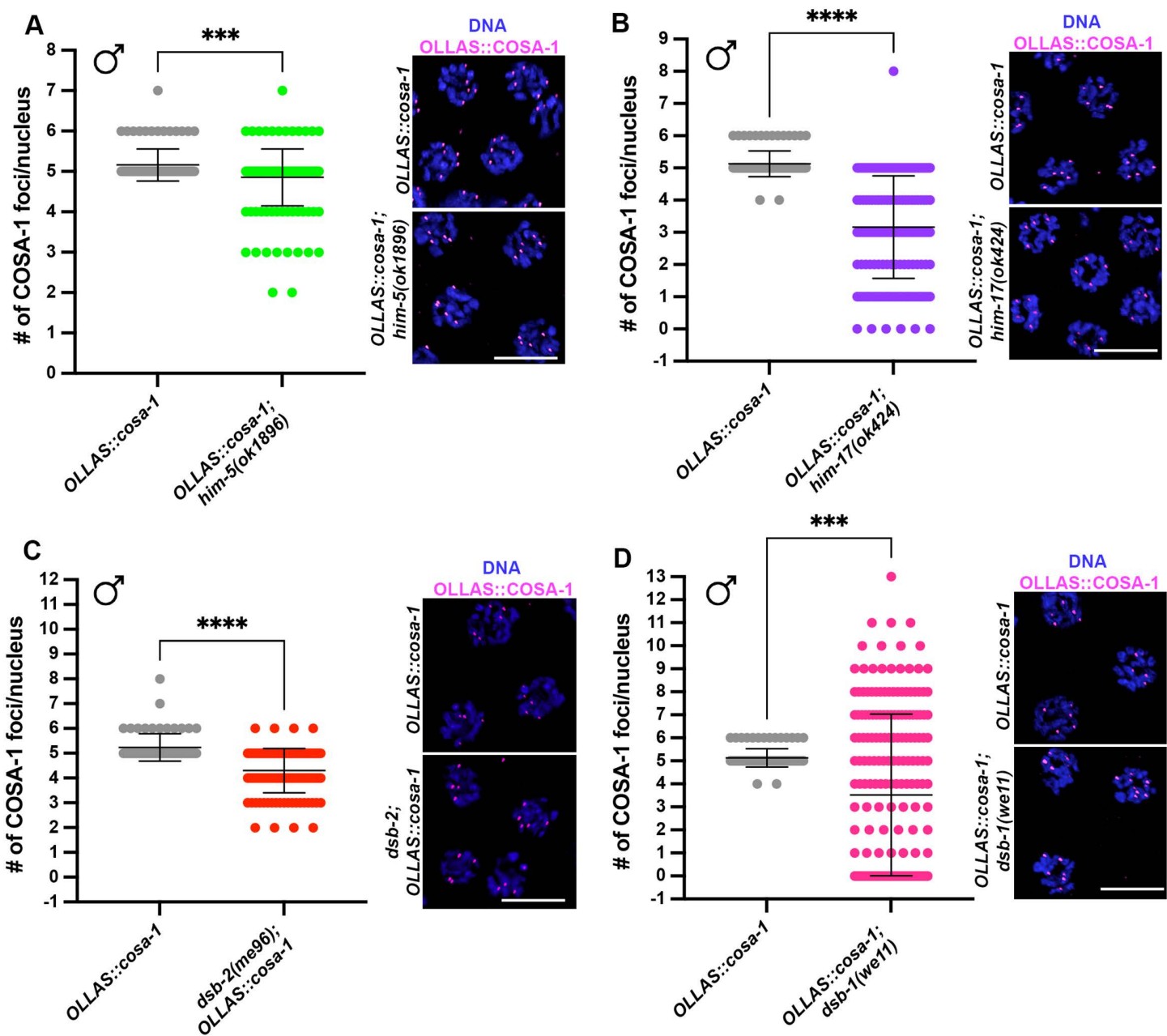

**Fig 5. Loss of DSB-1, but not HIM-5, HIM-17 or DSB-2 recapitulates SPO-11 deficiency for OLLAS::COSA-1 loading.** (A-D) Quantification and representative images of late pachytene cells stained for OLLAS::COSA-1 (magenta) and counterstained by DAPI (blue) in the indicated mutant backgrounds. Bars indicate mean with S.D. and asterisks denote statistical significance assessed by Kolmogorov-Smirnov test (****$p < 0.0001$). Scale bar 5 µm. The number of nuclei analyzed in controls and *him-5(ok1896)* worms was 93 and 158, respectively; in controls and *him-17(ok424)* worms was 93 and 169, respectively; in controls and *dsb-2(me96)* worms was 68 and 121, respectively; in controls and *dsb-1(we11)* worms was 93 and 212, respectively.

To that end, we constructed *top-1::AID::GFP; OLLAS::cosa-1*; *spo-11::AID* and *top-2::AID; OLLAS::cosa-1; spo-11::AID* worms carrying the SUN-1 promoter driven TIR1 (*ieSi38*) and examined COSA-1 in the presence and absence of auxin. We first assessed whether depletion of TOP-1 was occurring efficiently by monitoring GFP localization upon 24h exposure to auxin. We found that TOP-1::AID::GFP detection was largely abrogated, and RAD-51 foci were no longer detectable, suggesting that both TOP-1 and SPO-11 were efficiently depleted (S6A Fig). We assessed TOP-2 depletion by examining the mitotic region of the germ line and observed almost 100% enlarged and/or misshapen nuclei following auxin treatment for 24h (S6B Fig), as has been observed in *top-2* mutant alleles and indicating efficient depletion [59]. Quantification of COSA-1 foci following 24h of auxin treatment in both strains indicates that OLLAS::COSA-1 foci in late pachytene/condensation zone nuclei were still abundantly formed (Fig 6), suggesting that neither TOP-1 nor TOP-2 generate lesions in the absence of SPO-11 where COSA-1 accumulates. We noticed that under TOP-2 depletion conditions, the average number of COSA-1 foci was further enhanced compared to SPO-11 depletion only (Fig 6), which we partly attribute to the possible generation of polyploid cells arising from impaired cell division due to the loss of TOP-2 (S6B Fig).

### Sexual dimorphic role of SC component SYP-2 in promoting and limiting recruitment of COSA-1 upon SPO-11 depletion

In *C. elegans*, the SC is crucial for the successful formation of chiasmata. The tripartite structure of the SC is composed of lateral elements, consisting of a family of HORMA-domain containing proteins (HTP-3 [43,60], HIM-3 [61], and HTP-1/-2 [62,63]) and cohesins [64,65], and central elements, which polymerize between paired homologous chromosome axes. In worms, several central element components have been identified: SYP-1, -2, -3, -4, -5, -6 (SYPs) and SKR-1

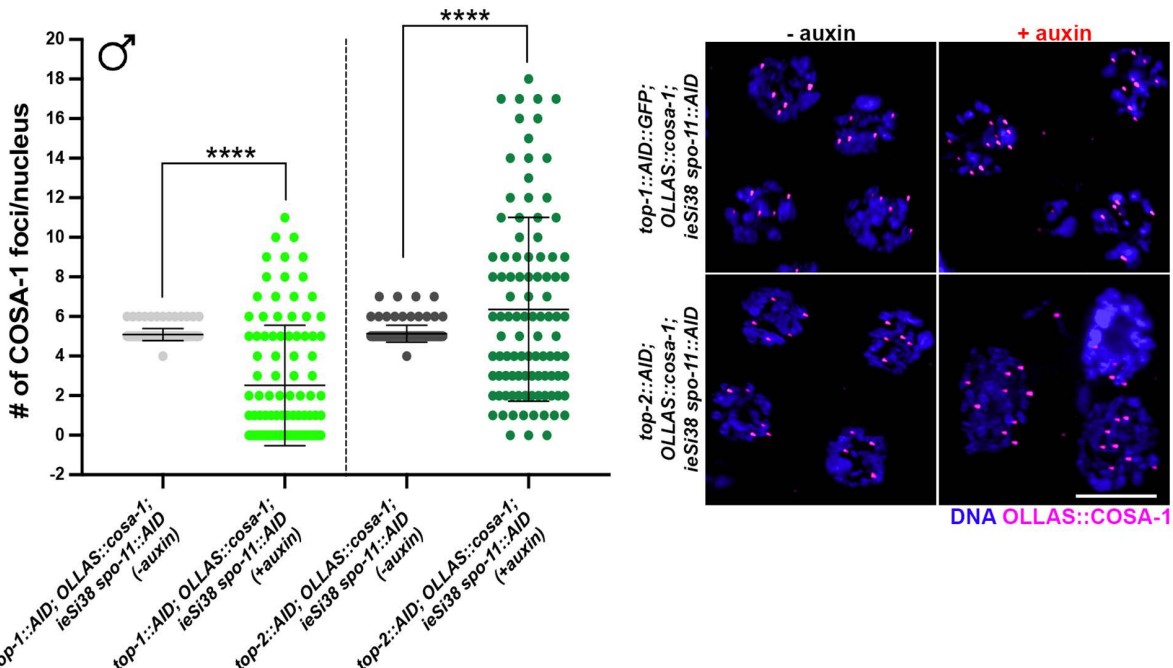

**Fig 6. HR-independent COSA-1 recruitment in males does not depend on TOP-1 or TOP-2.** Quantification (left) and representative images (right) of nuclei showing OLLAS::COSA-1 foci (magenta) counterstained with DAPI (blue) in the indicated genotypes and exposure conditions to auxin in late pachytene nuclei. Bars show mean with S.D. and asterisks denote statistical significance assessed by Kolmogorov-Smirnov test (****$p < 0.0001$, ns = *not significant*). The number of nuclei analyzed for *top-1::AID::GFP; OLLAS::cosa-1; spo-11::AID ieSi38* was 139 (-auxin) and 101 (+auxin). The number of nuclei analyzed for *top-2::AID; OLLAS::cosa-1; spo-11::AID ieSi38* was 138 (-auxin) and 100 (+auxin). Scale bar 5 μm.

and SKR-2, which all show interdependent loading [25,66–71]. Thus, abrogation of a single factor results in complete loss of synapsis. Growing evidence has shown that the SC can modulate recombination outcomes [7,72–74] and that recruitment/regulation of SYP proteins is sexually dimorphic, suggesting that the SC might possess different properties in the hermaphrodite and the male germ lines [10]. Previous work has shown that during oogenesis, DSBs are formed and accumulate in SYP mutants during meiotic progression, and substantial loading of pro-CO factors occurs in late pachytene cells [25,34,72,75]. However, diakinesis nuclei of synapsis-defective mutants mostly contain achiasmatic univalents, suggesting that the residual COSA-1 is likely to be recruited at intermediates that do not originate from inter-homolog recombination events.

We monitored COSA-1 loading in *spo-11::AID ieSi38; syp-2(ok307)* worms before and after auxin treatment in both male and hermaphrodite germ lines, and strikingly we observed opposing effects on COSA-1 foci formation in the two sexes. We found that loss of SYP-2 significantly reduced the number of SPO-11-independent COSA-1 foci in males (Fig 7A). In contrast, developing oocytes in the hermaphrodite gonad displayed a wide distribution, with a substantial cohort of nuclei bearing an elevated number of COSA-1 foci (Fig 7B), a situation never observed in hermaphrodites except under reduced (but not abrogated) synapsis [7]. To determine whether the elevated number of COSA-1 foci was due to incomplete depletion, we examined *spo-11(ok79); syp-2(ok307)* mutant hermaphrodites and observed elevated COSA-1 foci (S7A and S7B Fig). Importantly, we found that Diakinesis nuclei in the *spo-11(ok79); syp-2(ok307)* double mutants were indistinguishable from either *spo-11(ok79)* or *syp-2(ok307)* single mutants (S7C Fig), in which only achiasmatic univalents were found. This indicates that the recruitment of COSA-1 observed in the doubles does not sustain establishment of COs. Altogether, our findings unveil a differential role for SYP-2 (and likely the SC) in modulating the recruitment of pro-CO factors on recombination-independent lesions, where the number of COSA-1 foci is enhanced in the male germ line and reduced in the hermaphrodite germ line.

## Discussion

Here we explore how meiotic CO control is regulated in *C. elegans* males. We provide evidence that although males generate high quality sperm, they appear to have less robust CO control than hermaphrodites. Unexpectedly we find that COSA-1 and partners are enriched at genomic sites in late meiotic prophase in the absence of SPO-11, RAD-51, or Topoisomerases I or II, suggesting that CO markers bind to a previously unrecognized lesion/structure generated at the end of meiotic prophase in spermatogenesis.

### Germline sex-specific differences do not appear to influence COSA-1 numbers

Previous work has shown that the length of time spent in prophase, sex chromosome genotype, and the presence of germline apoptosis distinguish meiosis in *C. elegans* hermaphrodites and males. We show here that perturbing these properties to mimic the opposite sex does not alter the formation of extra COSA-1-marked events. This suggests that neither the shorter period spent in meiotic prophase, nor the absence of apoptosis, contributes to the class of extra COSA-1 foci observed in male spermatocytes. *X* chromosome genotype also does not appear to influence the formation of extra COSA-1 foci as we still observed a class of nuclei that contain an extra COSA-1 marked event in the sex determination mutant *tra-2*, where *XX* worms develop into males. Thus, even though the *X* chromosome of males lacks a pairing partner, it does not appear to be subject to the inter-chromosomal effect, whereby an unpaired genomic region leads to additional COs on paired chromosomes [33–35]. This is consistent with our previous findings that the single *X* chromosome present in males germ cells is not recognized as unpaired by the ataxia telangiectasia mutated (ATM) checkpoint kinase [22]. ATM plays a role in the establishment and maintenance of feedback mechanisms that ensure enough DSBs are induced to form a CO on each chromosome pair [76]. It is likely that transient pseudosynapsis, SC components aligned between *X* sister chromatids, and the unique chromatin environment shields the *X* of males from mechanisms that recognize its unpaired status [22].

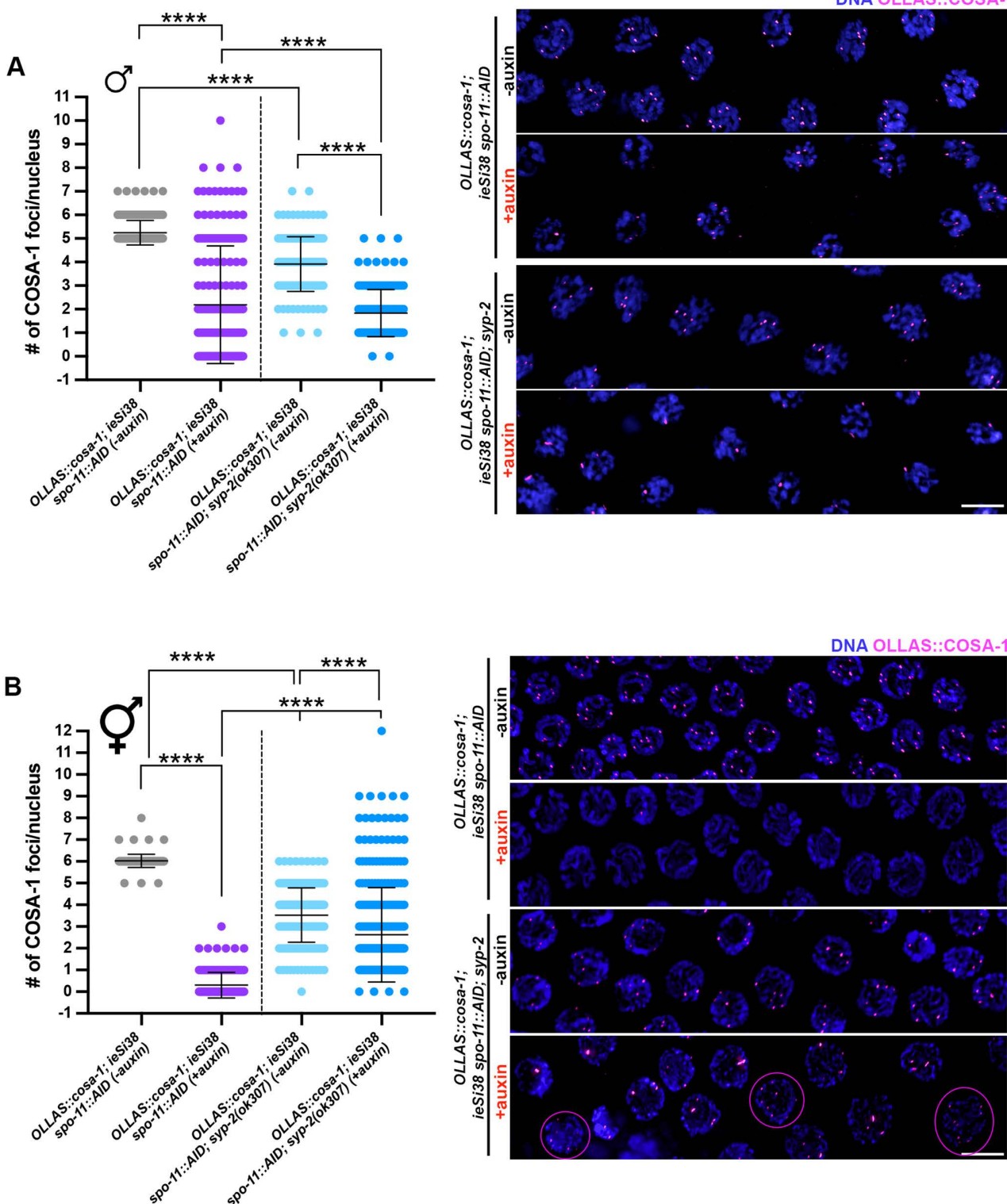

**Fig 7. Abrogation of synapsis exerts opposite effects on OLLAS::COSA-1 foci formation in male vs. hermaphrodite germ lines.** (A) Quantification and representative images of late pachytene cells in males of the indicated genetic background before and after exposure to auxin for 24h showing OLLAS::COSA-1 (magenta) counterstained with DAPI (blue). Bars indicate mean with S.D. and asterisks denote statistical significance assessed by

Kolmogorov-Smirnov test (****$p<0.0001$). Scale bar 5 µm. (B) Quantification and representative images of late pachytene cells in hermaphrodites of the indicated genetic background before and after exposure to auxin for 24h. Magenta circles indicate nuclei in which >6 COSA-1 foci can be observed. Bars indicate mean with S.D. and asterisks denote statistical significance assessed by Kolmogorov-Smirnov test (****$p<0.0001$). Scale bar 5 µm.

### Weakened CO homeostasis in spermatocytes

CO homeostasis ensures that each chromosome pair receives at least one CO, even under conditions of limiting DSBs. We show that unlike hermaphrodites, which efficiently convert limiting DSBs into COSA-1-marked events [28], male germ cells are less efficient at ensuring that limiting DSBs are processed into COSA-1-marked events (Fig 2). How then, with less robust CO homeostasis, do males ensure each homolog pair receives a CO? One possibility is that male germ cells induce more DSBs to ensure each homolog pair receives a CO. Consistent with this, RAD-51 foci accumulate to higher numbers in males compared to hermaphrodite germ cells [10,22,23]. Alternatively, or in addition to more DSBs, male spermatocytes may designate more DSBs into COs independently of COSA-1, or without accumulating COSA-1 into a cytologically detectable focus. In a variety of organisms, including yeast, mammals and plants, there are both class I COs, which are sensitive to interference, and class II COs, which are not subject to interference [3,4]. A few studies have provided evidence of non-interfering COs in *C. elegans* [77–82]. However, at least in some of these situations, the COs are dependent on COSA-1 but fail to accumulate cytologically detectable COSA-1 [28]. It is possible that males have a more robust class II CO pathway and/or are less likely to accumulate COSA-1 at CO sites, particularly under limiting DSBs.

### How do males produce high quality gametes when they lack key regulatory mechanisms?

A conundrum of investigations into sex-specific differences in *C. elegans* meiosis is that even though males appear to have less stringent meiotic regulation (e.g., undergo meiosis faster, have the unpaired *X* chromosome, lack germline apoptosis to cull defective germ cells, and do not have a robust system to both ensure and limit conversion of DSBs into COs), the fidelity of meiosis is higher in males compared to hermaphrodites [21]. This likely reflects the operation of more efficient mechanism(s) for segregation of chromosomes lacking a CO (achiasmate chromosomes) in males compared to hermaphrodites [83]. Sex differences in the ability to segregate achiasmatic chromosomes may be attributable to differences in the organization of chromosomes within nuclei at the end of meiotic prophase [17], and/or differences in the organization and assembly of meiotic spindles [84,85]. We also found previously that although checkpoints do not induce apoptosis in male germ cells, checkpoints are active and promote euploid sperm [21]. Alternatively, or in addition to achiasmatic chromosome segregation and robust checkpoint signaling, selection for high quality sperm may occur at fertilization.

### What is the nature of the COSA-1-marked sites in late meiotic prophase in *spo-11* mutants?

Surprisingly we found that COSA-1 and associated partners are present and show an altered distribution at chromosomal sites at late meiotic prophase in spermatogenesis when SPO-11 is not present. We show that these "sites" are not dependent on other topoisomerases, nor HR, suggesting some type of chromosomal lesion or alteration is present in late pachytene/condensation zone that attracts these CO markers when DSBs are not formed in male spermatocytes. Previous work demonstrated that upon transient heat shock, transposons are activated and lead to the formation of DSBs in the male, but not the hermaphrodite germ line [86]. It is unlikely that the COSA-1 foci observed in the absence of SPO-11 are due to transposon mobilization for the following reasons: one, transposon mobilization is dependent on heat shock but SPO-11-independent accumulation of COSA-1 is not; two, transposons are mobilized throughout the germ line, while COSA-1 foci in the *spo-11* mutant are only observed at very late pachytene into the condensation zone (Fig 3B and S1); three, RAD-51 accumulates at the sites of transposon activation but not in the absence of SPO-11 (Fig 3B); and four, transposon-induced breaks promote COs while SPO-11-independent lesions do not (S3D Fig) [86].

One possibility to explain the presence of pro-CO markers at the end of meiotic prophase is that during the process of condensation, critical for packaging the genome into sperm, there are breaks or distortions in the DNA that provide sites for accumulation of COSA-1 and other CO markers. These lesions/sites do not accumulate COSA-1 in wild-type meiosis because COSA-1 is enriched at *bona fide* COs. Previous studies have shown that in meiotic mutants that perturb CO formation, COSA-1 can bind DNA at sites other than at COs [24,34,72,75,87]. An alternative possibility is that in the absence of SPO-11 chromatin changes could leave genomic sites vulnerable to accumulation of these pro-CO factors. The broad distribution of COSA-1 foci we observe, instead of the five (or six) that mark COs in wild type, is consistent with these marking some type of stochastic lesion or site.

## Differential roles for DSB auxiliary proteins and the SC in males

In our analyses of SPO-11-independent accumulation of CO markers, we discovered different phenotypes caused by removal of DSB auxiliary proteins in males compared to hermaphrodites. COSA-1 accumulates predominantly at five foci in *him-5* hermaphrodites, presumably on the five autosomes but not the *X* chromosomes [56]. This is consistent with the activity of HIM-5 in specifically promoting DSBs for CO formation on the *X* chromosome. Since the single *X* chromosome of males does not accumulate COSA-1 [22], the significant decrease of foci that we observed (Fig 5A) suggests that HIM-5 has a more significant role in promoting DSBs on the autosomes in males. This could indicate that HIM-5 acts directly on autosomes to promote DSBs and/or that HIM-5 activity might be responsive to the X/autosome ratio and its absence in males leads to an overall reduction in DSBs.

In hermaphrodites, loss of both HIM-17 and DSB-2 has been shown to greatly reduce DSB formation and consequently the number of COSA-1 foci is severely impaired in these mutants [53,55]. However, our data show that during spermatogenesis, a large cohort of meiocytes achieves normal COSA-1 loading, indicating that loss of *him-17* or *dsb-2* has more subtle effects on DSB formation in males compared to hermaphrodites, or that redundancies between pro-DSB factors in males could be more robust than in hermaphrodites thus promoting sufficient (or nearly sufficient) DSBs even in the absence of these factors. Nonetheless, the finding that removal of DSB-1 in males, like removal of SPO-11, triggers accumulation of pro-CO markers in late Prophase I nuclei indicates that enrichment of pro-CO factors is caused by absence of DSBs *per se* rather than SPO-11 specifically. Consistent with this, irradiation of *spo-11* males blocks the accumulation of late pachytene COSA-1 foci associated with SPO-11 removal (S1A Fig). The essential role of DSB-1 in DSB formation is consistent with recent evidence that suggests that SPO-11 establishes a physical interaction specifically with DSB-1 (and likely with DSB-3), which in turn could promote recruitment of downstream factors [52]. The differential role of DSB proteins is also corroborated by phenotypic analyses that revealed more penetrant phenotypes in *dsb-1* and *dsb-3* mutants compared to *dsb-2*, in which DSB deficiency results in worsening of phenotypes in older animals [53], or to *him-17* and *him-5* nulls.

We also observed opposing effects caused by removal of the SC component SYP-2 on the accumulation of pro-CO factors in the absence of DSBs in males and hermaphrodites: while in males there was a reduction of COSA-1 foci in the absence of SYP-2, in hermaphrodites, SYP-2 removal led to more COSA-1 foci. We think this is likely due to the absence of the SC, as all SC components are interdependent and removing any one SC component completely blocks SC formation [25,66,67,69–71]. Perhaps holding chromosomes within the confines of the SC exposes sites for accumulation of pro-CO factors in males but prevents accumulation in hermaphrodites due to differences in chromatin structure. Previous work has shown that in the presence of SPO-11-mediated DSBs and reduced (but not abrogated) synapsis, CO interference is weakened in *C. elegans* oocytes, as extra numerary COSA-1 foci with reduced spacing are formed along the chromosome axes [7]. However, it is not known in worms whether modulating DSB number in the presence or absence of synapsis could trigger the same effect.

In conclusion, we have identified additional differences between hermaphrodite and male meiosis, including strength of CO homeostasis, consequence of removing different DSB auxiliary proteins, and SPO-11-independent accumulation of

pro-CO factors. While the identification of what these sites represent awaits future studies, our work identifies previously unappreciated aspects of male meiosis.

## Materials and methods

### Genetics

*C. elegans* strains used in this study are listed in S1 Table. Some nematode strains were provided by the Caenorhabditis Genetics Center, which is funded by the National Institutes of Health National Center for Research Resources (NIH NCRR). Strains were maintained on OP50 as food source and the experiments were performed at 20°C except as noted.

### CRISPR-mediated allele construction

*rad-51(xoe53)* stop-in and *top-2::aid* alleles were generated using the CRISPR-Cas9 ribonucleoprotein complexes based on the co-CRISPR method [88]. A 113 base DNA oligonucleotide (IDT: 5'-GCGTGCAGCTGATCAAGCTTTGCTCAATGCAGCAAGGGAAGTTT GTCCAGAGCAGAGGTGACTAAGTGATAAGCTAGCTCGAGGATAATGCGATGGAGCAAGATGAAAACTTG-3') was used as the repair template for *rad-51(xoe53)*, which contains stop codons in all three reading frames. Briefly, the Cas9-crRNA-tracrRNA ribonucleoprotein complex (guide RNA: 5'-TTTGCTGAATGCAGCAATCG-3'), along with the repair template, was microinjected into the *C. elegans* gonad. F1 progenies exhibiting roller/dumpy phenotypes were isolated and genotyped by PCR (forward: 5'-GCAAGTCGTCAAAAGAAATCGG-3'; reverse: 5'-CCTGAAGACTCAAGTTTATCG-3') to confirm insertion (WT = 131 bp; *xoe53* = 213 bp). The guide RNA for the *top-2::aid* allele was: ACGCGTCGTCGACTCCGACT (plus strand, cuts 8 bps upstream of STOP codon) and the repair oligos were F:gccaaagaagaagaggggacgcgtcgtcgactccgaTtcAgaCatgcctaaagatccagccaaacctccggccaaggcacaagttgtgggatggccaccggtgagatcataccggaagaac and R: tgggatggccaccggtgagatcataccggaagaacgt gatggtttcctgccaaaaatcaagcggtggcccggaggcggcggcgttcgtgaagtaaattaatttgtttcccaccttccttaagtgttt.

Functional *GFP::rmh-1* tagged line was generated by CRISPR-Cas9 by Suny Biotech (https://www.suny-biotech.com/) and the genotyping was performed by using primers 5´-CGTACCTGCACATCAAACAT-3´and 5´-GTTCACCAATCTGCTTCACA-3´. Functional *cdk-2::HA* tagged line was previously characterized [31]. All strains were outcrossed for a minimum of three times before analyses.

### Irradiation time course

γ irradiation was performed at 24h post-L4 using a Cs-137 source and worms were dissected 8hr after exposure to various IR doses.

### Embryonic lethality of male-sired progeny

A single *fog-2* female was placed on individual plates with a male and allowed to mate and lay eggs. After 24hrs, they were transferred to new plates and this process was repeated for 3 days. Embryonic lethality was determined by counting eggs and hatched larvae 24hrs after removing the adult female and percent was calculated as eggs/(eggs + larvae).

### Cytological analyses

Immunostaining of germ lines was performed as described [18] except slides were incubated in 100% ethanol instead of 100% methanol for detection of direct fluorescence of GFP::COSA-1. Primary antibodies are listed in S2 Table. Life Technologies secondary donkey anti-rabbit antibodies conjugated to Alexa Fluor 488 or 594 and goat anti-rabbit conjugated to Alexa Fluor 647 were used at 1:500 dilutions. DAPI (2μg/ml; Sigma-Aldrich) was used to counterstain DNA.

Collection of fixed images was performed using an API Delta Vision Ultra deconvolution microscope or with a fully motorized widefield upright microscope Zeiss AxioImager Z2, equipped with a monochromatic camera Hamamatsu ORCA Fusion, sCMOS sensor, 2304 x 2304 pixels, 6.5 x 6.5 µm size. Z-stacks were set at 0.2 mm thickness and images were deconvolved using Applied Precision SoftWoRx batch deconvolution software or with ZEN Blue Software using the "constrained iterative" algorithm set at maximum strength. Whole projections of deconvolved images were generated with Fiji (ImageJ) (Wayne Rasband, NIH) and processed in Photoshop, where some false coloring was applied.

For RAD-51 quantification, germ lines were divided into the transition zone (leptotene/zygotene, from the first to last row with two or more crescent-shaped nuclei), and pachytene (divided into 3 equal parts: early, mid, and late pachytene). RAD-51 foci per nucleus was scored from half projections of the germ lines for each divided region. GFP::COSA-1 or OLLAS::COSA-1 foci were scored from deconvolved 3D z-stacks in mid-late pachytene nuclei individually to ensure that all foci within each individual nucleus were counted. Quantification of meiotic progression was calculated based on the nuclear row position of the most proximal EdU-positive nuclei where at least 50% of the nuclei in the row were EdU-positive; 4 germ lines were quantified.

## Auxin-induced degradation

A 440 mM stock solution of 3-Indoleacetic acid (Auxin) in Ethanol was freshly prepared every two weeks and dissolved into NG medium at a final concentration of 1 mM. Auxin-containing plates were kept refrigerated, protected from light, and used within three weeks. All experiments conducted on Auxin plates were set by selecting L4 males or hermaphrodites, and worms were dissected after 24h (or as indicated).

## Supporting information

**S1 Fig. SPO-11-independent GFP::COSA-1 foci.** (A) Images of pachytene/condensation zones of male [*meIs8[unc-119(+) pie-1promoter::GFP::cosa-1]; spo-11(ok79)* germ lines imaged for GFP::COSA-1 fluorescence (green) and counterstained with DAPI (blue) in the absence (0Gys) or presence (100Gys) of IR. Bracketed region denotes faint GFP::COSA-1 foci in the [*meIs8[unc-119(+) pie-1promoter::GFP::cosa-1]; spo-11(ok79)* mutant in the absence of IR. (B) Image of pachytene/condensation zones of *GFP::cosa-1(xoe44); spo-11(ok79)* males, L4 hermaphrodites undergoing spermatogenesis and adult hermaphrodites (24h post L4). White arrow denotes direction of meiotic progression. Bracket denotes region of germ line with GFP::COSA-1 foci. Scale bar 10 µm. (C) Stacked bar graph showing percent nuclei with indicated numbers of GFP::COSA-1 foci in *GFP::cosa-1(xoe44)*; *spo-11(ok79)* males, L4 and adult hermaphrodites. Three germ lines were examined; 80 nuclei were scored in males, 58 nuclei were scored in L4 hermaphrodites and 98 nuclei were scored in adult hermaphrodites. Number of GFP::COSA-1 foci: grey = 0; blue = 1; red = 2; purple = 3; neon green = 4; cyan = 5; rose = 6; gold = 7. (D) Mean number and S.D. of GFP::COSA-1 as scored through the germ line from early pachytene (EP), mid pachytene (MP) and late pachytene (LP) from a minimum of 3 germ lines.
(TIF)

**S2 Fig. RMH-1 is recruited at early and late meiotic stages in the male germ line.** Whole-mount male germ line of the indicated genotype immunolabeled for GFP::RMH-1 (green) and OLLAS::COSA-1 (red) counterstained with DAPI (blue), showing abundant RMH-1 foci formation in early pachytene and recruitment to presumptive CO sites together with COSA-1 in late pachytene cells. PMT: pre-meiotic tip; TZ: Transition Zone; EP: Early Pachytene; LP: Late Pachytene. Scale bar 20 µm.
(TIF)

**S3 Fig. The newly generated *xoe53* is a putative *rad-51* null allele.** (A) Top: Whole-mount hermaphrodite gonads of the indicated genetic backgrounds immunoassayed for RAD-51 (red) and counterstained with DAPI (blue). Scale bar 20

μm. Color-coded insets (bottom left) show magnified early-mid pachytene regions to show no detectable RAD-51 foci in the *OLLAS::cosa-1; spo-11::AID rad-51(xoe53) ieSi38* worms compared to *OLLAS::cosa-1; spo-11::AID ieSi38* controls. Scale bar 20 μm. Bottom right: representative images of DAPI-stained diakinesis nuclei depicting aberrant chromatin bodies in the *OLLAS::cosa-1; spo-11::AID rad-51(xoe53) ieSi38* (-auxin) whose formation is suppressed upon SPO-11 depletion (+auxin). Scale bar 2 μm. (B) Left: whole-mount male gonads of the indicated genetic backgrounds immunolabeled for RAD-51 (red) and counterstained for DAPI (blue). Scale bar 20 μm. Right: magnified color-coded insets showing early-pachytene nuclei in control and *rad-51(xoe53)* mutant worms showing no detectable RAD-51 foci. Scale bar 2 μm. (C) Top: late-pachytene oocytes in the indicated genetic backgrounds and exposure conditions to auxin immunolabeled for OLLAS::COSA-1 (red) and counterstained for DAPI (cyan). Scale bar 5 μm. Bottom: quantification of OLLAS::COSA-1 foci number in the same genotypes and exposure conditions to auxin. Bars show mean with S.D. and asterisks denote statistical significance assessed by Kolmogorov-Smirnov test (****$p < 0.0001$, ns = not significant). The number of nuclei analyzed in controls and *rad-51(xoe53)* are 122–172 (-auxin) and 130–160 (+auxin). (D) Percent inviability of *fog-2(q71)* progeny sired by WT (N2), *spo-11(ok79)*, and *cosa-1(tm3298)* males. Mean and 95% confidence intervals are shown from a total of 8 matings. No statistical difference between *spo-11* and *cosa-1* was observed by Mann-Whitney.
(TIF)

**S4 Fig.  Removal of pro-DSB accessory factors results in milder effects on RAD-51 accumulation in males.** (A) Quantification (top) and representative images (bottom) of nuclei from controls, *him-5* and *him-17* mutants at the indicated stages immunoassayed for RAD-51 (red) and counterstained with DAPI (blue). Scale bar 5 μm. The number of nuclei analyzed from zone 1–6 was: *OLLAS::cosa-1* (86 – 63 – 53 – 48 – 47 – 40), *OLLAS::cosa-1; him-5(ok1896)* (129 – 70 – 56 – 59 – 61 – 65), *OLLAS::cosa-1; him-17(ok424)* (104 – 66 – 44 – 53 – 51 – 62). (B) Quantification (top) and representative images (bottom) of nuclei from aged (48h post L4) controls and *dsb-2* mutants at the indicated stages immunoassayed for RAD-51 (red) and counterstained with DAPI (blue). Scale bar 5 μm. The number of nuclei analyzed from zone 1–6 was: *OLLAS::cosa-1* (54 – 28 – 21 – 21 – 21 – 25), *OLLAS::cosa-1; dsb-2(me96)* (77 – 72 – 49 – 44 – 43 – 52).
(TIF)

**S5 Fig.  Recombination defects observed in *dsb-2* mutants are more severe in hermaphrodites than males.** (A) Whole-mount hermaphrodite and male germ lines dissected 48h post-L4 and immunostained for RAD-51 (red)/ OLLAS::COSA-1 (green) and counterstained with DAPI (blue). Scale bar 20 μm. (B) Color-coded insets showing magnified early and late pachytene cells from the indicated sexes depicting more robust recruitment of RAD-51 and COSA-1 in males versus hermaphrodites. Scale bar 5 μm.
(TIF)

**S6 Fig.  TOP-1 and TOP-2 are efficiently depleted upon exposure to auxin.** (A) Whole-mount male germ lines from the indicated genotype immunoassayed for TOP-1::AID::GFP (green), RAD-51 (red) and counterstained with DAPI (blue). TOP-1 is efficiently depleted upon 24h exposure to auxin. Scale bar 20 μm. (B) DAPI staining of the distal tip from *top-2::AID* animals before and after exposure to auxin. Note mitotic catastrophe elicited by TOP-2 depletion, as indicated by enlarged nuclei, micronuclei and chromatin bridges. Scale bar 2 μm.
(TIF)

**S7 Fig.  Loss of SYP-2 triggers accumulation of COSA-1 foci in *spo-11* mutant hermaphrodites.** (A) Left: representative images of late pachytene oocytes of the indicated genotypes stained with OLLAS (COSA-1; magenta) and counterstained with DAPI (blue). Scale bar 2 μm. Right: quantification of OLLAS::COSA-1 foci in the indicated mutant backgrounds. The number of nuclei analyzed was: *OLLAS::cosa-1* (116), *OLLAS::cosa-1; syp-2(ok307)* (123), *OLLAS::cosa-1; spo-11(ok79)* (180), *OLLAS::cosa-1; spo-11(ok79); syp-2(ok307)* (426). Bars indicate S.D. and asterisks denote statistical significance assessed by Kolmogorov-Smirnov test (****$p < 0.0001$). (B) High magnification of

two examples of late pachytene nuclei from *OLLAS::cosa-1; spo-11(ok79); syp-2(ok307)* mutants displaying extensive OLLAS::COSA-1 foci. Scale bar 2 μm. (C) Quantification (top) and representative images (bottom) of DAPI bodies in Diakinesis nuclei of the indicated genotypes. Scale bar 2 μm. Bars depict S.D. and statistical comparison by T test indicates non-significant difference (*ns*). The number of Diakinesis nuclei scored was: *OLLAS::cosa-1* (41), *OLLAS::cosa-1; syp-2(ok307)* (63), *OLLAS::cosa-1; spo-11(ok79)* (64), *OLLAS::cosa-1; spo-11(ok79); syp-2(ok307)* (63).
(TIF)

**S1 Table.  Worm strains used in this study.**
(DOCX)

**S2 Table.  Antibodies used in this study.**
(DOCX)

**S1 Data.  Primary data for Figs 1, 2, and S1.**
(XLSX)

## Acknowledgments

We are grateful to Yumi Kim and Sevinc Ercan for strains. We also thank Daniel Elnatan for thoughtful discussion and help with image analyses. Some strains were provided by the CGC, which is funded by NIH Office of Research Infrastructure Programs (P40 OD010440). We acknowledge the core facility CELLIM supported by MEYS CR (LM2023050 Czech-BioImaging) and Biological Data Management and Analysis Core Facility funded by ELIXIR CZ research infrastructure (MEYS Grant No: LM2023055). We thank the MCB Light Microscopy Imaging Facility, which is a UC Davis Campus Core Research Facility, for the use of the Deltavision Ultra microscope for generating images.

## Author contributions

**Conceptualization:** JoAnne Engebrecht, Nicola Silva.

**Funding acquisition:** JoAnne Engebrecht, Enrique Martinez-Perez, Nicola Silva.

**Investigation:** JoAnne Engebrecht, Aashna Calidas, Qianyan Li, Angel Ruiz, Pranav Padture, Neeraj Bhavani Aniyan Bhavana, Consuleo Barroso, Nicola Silva.

**Methodology:** Qianyan Li.

**Supervision:** JoAnne Engebrecht, Enrique Martinez-Perez, Nicola Silva.

**Visualization:** JoAnne Engebrecht, Aashna Calidas, Nicola Silva.

**Writing – original draft:** JoAnne Engebrecht, Nicola Silva.

**Writing – review & editing:** JoAnne Engebrecht, Aashna Calidas, Qianyan Li, Angel Ruiz, Pranav Padture, Neeraj Bhavani Aniyan Bhavana, Consuleo Barroso, Enrique Martinez-Perez, Nicola Silva.

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
