## [Decision Letter · Decision Letter 0]

27 Jul 2025

PGENETICS-D-25-00694

Loss of meiotic double strand breaks triggers recruitment of recombination-independent pro-crossover factors in C. elegans spermatogenesis

PLOS Genetics

Dear Dr. Engebrecht,

Thank you for submitting your manuscript to PLOS Genetics. After careful consideration, we feel that it has merit but does not fully meet PLOS Genetics's publication criteria as it currently stands. Therefore, we invite you to submit a revised version of the manuscript that addresses the points raised during the review process.

Please submit your revised manuscript within 30 days Aug 26 2025 11:59PM. If you will need more time than this to complete your revisions, please reply to this message or contact the journal office at plosgenetics@plos.org. Please include the following items when submitting your revised manuscript:

We look forward to receiving your revised manuscript.

Kind regards,

Aimee Jaramillo-Lambert, Ph.D.

Guest Editor

PLOS Genetics

Monica Colaiácovo

Section Editor

PLOS Genetics

Aimée Dudley

Editor-in-Chief

PLOS Genetics

Anne Goriely

Editor-in-Chief

PLOS Genetics

**Additional Editor Comments:**

Guest Editor comments:

I share the questions of reviewers 2 and 3 regarding the altered distribution of COSA-1 in the SPO-11 depleted germ lines. It would be helpful to discuss why more COSA-1 foci would be present and why there is such a broad distribution of COSA-1 foci.

Throughout the manuscript and figures TIR is referenced in several different ways (e.g. TIR1, TIR1::mRuby, or not at all) making it a little confusing as to when it is present and whether there are different TIR1 genes present. Please edit for consistency and clarity.

Figure 5. Related to comment 6 from reviewer 3; Provide the genotype for the strain(s) used in Panel C or add the proteins to GFP and OLLAS.

Figure 7. Please include the statistics for OLLAS::cosa-1; spa-11::AID TIR -auxin and plus auxin (Panel A) and between OLLAS::cosa-1; spo-11::AID TIR1 (-auxin) and OLLAS::cosa-1; spo-11::AID TIR1 syp-2(ok307) (-auxin) (Panel B).

In the section "SPO-11-independent COSA-1 foci also form independently of Topoisomerase I or II," care is taken to show that TOP-1 is depleted efficiently, but there is no mention of how TOP-2 depletion was monitored. Please include.

**Journal Requirements:**

1) We notice that your supplementary Tables are included in the manuscript file. Please remove them and upload them with the file type 'Supporting Information'. Please ensure that each Supporting Information file has a legend listed in the manuscript after the references list.

2) Some material included in your submission may be copyrighted. According to PLOSu2019s copyright policy, authors who use figures or other material (e.g., graphics, clipart, maps) from another author or copyright holder must demonstrate or obtain permission to publish this material under the Creative Commons Attribution 4.0 International (CC BY 4.0) License used by PLOS journals. Please closely review the details of PLOSu2019s copyright requirements here: PLOS Licenses and Copyright. If you need to request permissions from a copyright holder, you may use PLOS's Copyright Content Permission form.

Potential Copyright Issues:

i) Figure 1. Please confirm whether you drew the images / clip-art within the figure panels by hand. If you did not draw the images, please provide (a) a link to the source of the images or icons and their license / terms of use; or (b) written permission from the copyright holder to publish the images or icons under our CC BY 4.0 license. Alternatively, you may replace the images with open source alternatives. See these open source resources you may use to replace images / clip-art:

3) We note that your Data Availability Statement is currently as follows: "All relevant data are within the manuscript and its Supporting Information files.". Please confirm at this time whether or not your submission contains all raw data required to replicate the results of your study. Authors must share the “minimal data set” for their submission. PLOS defines the minimal data set to consist of the data required to replicate all study findings reported in the article, as well as related metadata and methods (https://journals.plos.org/plosone/s/data-availability#loc-minimal-data-set-definition).

4)  Please ensure that the funders and grant numbers match between the Financial Disclosure field and the Funding Information tab in your submission form. Note that the funders must be provided in the same order in both places as well.  

5) Please send a completed 'Competing Interests' statement, including any COIs declared by your co-authors. If you have no competing interests to declare, please state "The authors have declared that no competing interests exist". Otherwise please declare all competing interests beginning with the statement "I have read the journal's policy and the authors of this manuscript have the following competing interests"

**Reviewers' comments:**

Reviewer's Responses to Questions

**Comments to the Authors:**

Reviewer #1: The study by Engebrecht et al. reports ways in which a pro-crossover factor, COSA-1, exhibits distinctively different behavior in male germline nuclei relative to hermaphrodite (oogenic) germline nuclei; the distinctive COSA-1 behavior anchors multiple experimental directions that reveal new and interesting information about male germline nuclei.

The manuscript begins with a relatively short section reporting a difference in the number of observed versus expected COSA-1 foci in the context of SPO-11+ male versus hermaphrodite germlines; namely, an additional COSA-1 focus is found in ~5% of prophase nuclei from male germlines. This first part of the study thus reports findings that may suggest the C. elegans male germline has “less robust crossover control” (as stated in the Abstract), so long as we are sure that every COSA-1 focus reliably becomes a crossover.

The second part of the study (most closely relating to the Title and majority of experiments) explores the unexpected finding that - when germlines are devoid of SPO-11 activity (whose DNA double strand break activity initiates meiotic recombination) - multiple COSA-1 foci localize to chromatin in a substantial fraction of late prophase nuclei. This phenomenon appears to occur in both male and hermaphrodite germlines (according to my eye, see below) but is more extensive (a greater number of COSA-1 foci per nucleus are found) in the male germline. The finding suggests that SPO-11 activity normally prevents the accumulation, in late prophase, of a substrate to which the COSA-1 pro-crossover factor (and co-factors) bind. The authors’ find that RAD-51 is not involved in the production of these late prophase COSA-1 substrates caused by SPO-11 removal, and that Topoisomerase II acts, to some extent, to prevent them (Fig. S5) but SYP-2 promotes their existence (at least in males; in hermaphrodites SYP-2 seems to prevent them!).

The authors thoroughly analyze the role of several meiotic factors in each of the novel events reported here. But the study lacks a degree of cohesiveness, I think because of the remaining unknowns and loose connection between the two (SPO-11+ and spo-11 mutant) phenomena described. An example of a remaining unknown is (I think) whether the extra COSA-1 focus observed in SPO-11+ male germline nuclei corresponds to an interhomolog crossover event. If not, then the presence of this extra COSA-1 focus perhaps is not a cytological reflection of less robust crossover control in the male germline. (Prior published genetic data indicating more double crossovers in the male does indicate that the male germline has less stringent crossover control, I am only questioning the rigor of interpreting this extra COSA-1 focus in the male germline as a crossover, particularly when the second part of this study focuses on COSA-1 foci that are apparently unassociated with crossovers and tend to emerge in the male germline.) Second, the molecular basis for COSA-1 foci that arise when SPO-11 is depleted remains unclear. These COSA-1 foci are not apparently connected to “normal” recombination, but knowing what molecular entity it binds to could reveal interesting information about the substrate COSA-1 prefers, which could address how it functions in normal recombination. Finally, it is not clear whether this phenomenon (appearance of COSA-1 foci when SPO-11 is absent) is connected in any way to the extra COSA-1 focus occasionally observed in SPO-11+ male germline nuclei.

I have several suggestions that may help streamline the narrative to improve its coherence and/or readability. There are also a couple of places where an additional experiment (or, if experimentation not possible, then revision to the narrative) would improve the rigor and/or completeness of the study. Illustrations of models to help the reader synthesize the different threads of the paper might be useful, particularly one that helps the reader understand how SYP-2 might attenuate COSA-1-associated lesions in hermaphrodites but promote them in the male germline.

Comments

1. Title: The phrase “recombination-independent pro-crossover factors” seems counterintuitive - a pro-crossover factor cannot be independent of recombination (since crossovers are, by definition, recombination events). Perhaps better to state that the recruitment is recombination-independent instead of the factors themselves being recombination-independent. Perhaps: “Loss of meiotic double strand breaks triggers recruitment of pro-crossover factors to chromatin in C. elegans spermatogenesis”

2. I agree that prior genetic data indicating more double crossover events provides evidence for less robust crossover control in the male. However I am less convinced that the extra COSA-1 focus observed in the male germline is a cytological reflection of less robust crossover control (perhaps I am too influenced by the second part of the study that demonstrates additional recombination-independent substrates for COSA-1 arise, particularly in male germlines). This is just a comment - In the end I do not think this is an issue needing resolved for this study, as the extra COSA-1 foci in the male germline is a difference between spermatogenesis/oogenesis worth exploring regardless of whether it corresponds to a future crossover or not. This uncertainty however (in my brain) tends to muddle my thinking of whether the SPO-11+ and spo-11 minus COSA-1 phenomena are somehow connected.

3. Abstract: “and that there are previously unrecognized lesions or structures at the end of meiotic prophase in spermatocytes that can accumulate CO markers” And, at the end of the Introduction: "Surprisingly, we discovered that COSA-1 and its partners accumulate at the end of meiotic prophase in the absence of DSBs specifically in spermatogenesis." But don’t hermaphrodite germlines show the same lesions, only fewer of them per nucleus? (Supplemental Fig 7 indicates that one to three COSA-1 foci are observed in a substantial fraction of spo-11 hermaphrodite germline nuclei.) If I understand this correctly, the Abstract and Intro should be modified to avoid implying that COSA-1 associated lesions arising due to SPO-11 removal are exclusive to the male germline.

4. Currently the study reads almost like it is two separate papers back-to-back. To streamline the narrative, I recommend that data in Figures 1 and 2 be consolidated (the genetic pathways in Figure 2 are not necessary and can be removed). The Results describing experiments in Figures 1 and 2 could maybe also be made more concise. If the first part (addressing COSA-1 foci in SPO-11+ germlines) is shorter, the reader may be more attentive to the distinctive aspects of the second set of major observations (which I think make up the majority of the study).

5. Using COSA-1 as a marker for crossovers and gamma radiation to introduce DNA DSBs, the authors observe that males have a (more or less) linear response in converting increasing #s of DSBs into crossovers. The narrative concludes that males are different from hermaphrodites in this respect, but only shows a gamma radiation dose experiment performed on males, comparing the results to prior hermaphrodite data from a different lab. For maximal confidence in the conclusion that males and hermaphrodites differ in their degree of crossover homeostasis at low and high doses of radiation, hermaphrodites and males should be treated in parallel (optimally exposed to radiation in the same culture dish) and using the same equipment and immunofluorescence conditions. If this kind of rigorous comparison is not possible, the strength of conclusions made (about differences between male and hermaphrodites) should be dialed back in the narrative.

6. Figure 4’s title reads “SPO-11 activity in males is extended throughout prophase.” But the data shown in this figure, suggest (to me) that SPO-11 activity is somewhat restricted to zones 1 and 2, not extended throughout prophase. Even after 2 hours of treatment with auxin to remove SPO-11 activity, RAD-51 foci only substantially diminish in zone 1 and 2 (not 3 and 4). Only after 4 hours of exposure to auxin do we observe substantive diminishment of RAD-51 in the later zones 3 and 4. I suggest changing the title, and perhaps instead focus the title on the surprise finding that SPO-11 removal causes recruitment of COSA-1 foci to chromatin.

7. The Results section “SPO-11 depletion in the male germ line leads to a similar abrogation of DSBs as in the hermaphrodite germline” is written clearly, but nevertheless I read it a few times because I was unsure what question the authors were addressing with the experiment. The end of the section states “These results indicate that depletion of SPO-11 leads to reduction of DSBs in a similar pattern to hermaphrodites”, but it is unclear what this experiment was trying to get at in the first place. In other words, was there a different possible outcome that would have suggested something specific about recombination in the male germline? Because this section seems to be an experiment performed without a specific hypothesis in mind, I wonder if this section subheading could be removed and these results streamlined/combined with the following section that reports the interesting observation of [COSA-1 foci induced by SPO-11 removal]? The data on RAD-51 and fertility could perhaps be extracted and constitute an independent Results subsection.

7. Results subheading: “COSA-1 foci accumulate independently of SPO-11 and RAD-51 in late pachytene spermatocytes” – this heading is technically correct but caution to the authors because it might imply that there is a population of SPO-11 and RAD-51-independent COSA-1 foci even in normal wild type meiosis. An alternative title could be: “Removal of SPO-11 causes the appearance of COSA-1 foci in late pachytene” (or, “RAD-51-independent COSA-1 foci in late pachytene”, if the authors decide not to extract the RAD-51 data into an independent results subsection).

8. In the Results subsection mentioned above (pt. 7), the authors imply that hermaphrodites depleted of SPO-11 have a block in COSA-1 formation, and a prior study is cited. However, data in Figure 7 (to my eyes) indicates that many nuclei in the SPO-11-depleted hermaphrodite germline have one, two, or three COSA-1 foci. This set of hermaphrodite data (COSA-1 foci in SPO-11 depleted hermaphrodite germlines in Figure 7) should be brought to this earlier position of the manuscript and fully discussed for clear comparison between males and hermaphrodites. (It is not clear to me what region of the hermaphrodite gonad these COSA-1 foci-positive nuclei are found, thus I think this also should be indicated, so that the reader can understand how they may or may not connect to the COSA-1 positive nuclei found in SPO-11- depleted male germlines.) The data suggest (to me) that COSA-1 foci are found in some hermaphrodite germline nuclei, although they do not rise to the abundance found in some male germline nuclei. Male germline nuclei in the (spo-11 knockout) hermaphrodite L4, however, seem to exhibit predominantly 1-3 COSA-1 foci, resembling what appears to be the case for hermaphrodite oogenic germline nuclei. If I have understood the data correctly, the statements implying that COSA-1 – positive nuclei are specific to male germline nuclei should be modified (e.g.: “and is specific to spermatogenesis in both males and L4 hermaphrodites.”)

9. The authors raise an interesting prior observation that “progeny sired by spo-11(ok79) males have a higher viability than progeny derived from spo-11 mothers [40]” and raise the possibility that perhaps this improved fertility of spo-11 males is due to COSA-1 mediated CO formation. The rigorous experiment to address this question would be to remove COSA-1 from spo-11(ok79) males (i.e. create the cosa-1 spo-11 double mutant) and evaluate the viability of progeny sired. Instead, the viability of progeny sired by cosa-1 single mutant males is compared to the viability of progeny sired by spo-11 mutant males. A rigorous conclusion cannot be drawn about whether COSA-1 crossovers (or crossover-like entities) promote positive chromosome disjunction outcomes in spo-11 mutant males from the single mutant comparisons: There is the possibility that SPO-11-mediated (but COSA-1 independent) recombination intermediates promote positive chromosome disjunction outcomes in the cosa-1 single mutant and likewise that COSA-1 associated entities promote positive chromosome disjunction outcomes in the spo-11 single mutant (resulting in similar fertility between the two male genotypes). If the spo-11 cosa-1 double mutant cannot be analyzed (perhaps these genes are very tightly linked), the narrative should be revised to discuss these limitations to the interpretation.

9b. (this is more of an aside): Could it be the case that spo-11 hermaphrodites cull germline nuclei with high numbers of COSA-1 foci? I seem to recall that spo-11 hermaphrodites display little germ cell apoptosis, but thought I should ask in case it is a possibility.

10. Another suggestion for manuscript readability is to move the paragraph starting “To determine if other CO markers are also enriched with COSA-1 in the absence of SPO-11…” earlier, prior to the fertility data and RAD-51 co-localization / dependency experiments. These “COSA-1 partner” data naturally follow the original observation of COSA-1 foci, prior to conducting dependency experiments (which only analyze COSA-1). I do think the RAD-51 colocalization and dependency experiments would be best presented as their own independent Results subsection.

11. I suggest limiting the interpretation of the SYP-2 experiments somewhat in the narrative (Results subtitle and some words throughout the manuscript that suggest SC is the only thing disrupted in a syp-2 mutant): If only SYP-2 is analyzed, it is most rigorous to conclude that a particular outcome reflects a role for SYP-2 (which may perhaps reflect a role for the SC). SYP-2 could possibly have an SC-independent role in promoting these lesions when SPO-11 is removed; more than one SC component should be analyzed if one wants to feel confident in attributing an outcome to lack of SC itself. So for example I suggest revising the Results subheading to “Sexual dimorphic role of SYP-2 in promoting and limiting recruitment of COSA-1 upon SPO-11 depletion” (or “…SC component SYP-2…”) and revising sentence “We found that loss of synapsis significantly reduced the number of SPO-11-independent COSA-1 foci in males” to “We found that loss of SYP-2 significantly reduced the number of SPO-11-depletion-associated COSA-1 foci in males”, and the final sentence to “Altogether, our findings unveil a differential role for SYP-2 (and possibly the SC) in modulating the recruitment of…”

12. “However, diakinesis nuclei of synapsis-defective mutants mostly contain achiasmatic univalent, suggesting that the residual COSA-1 is likely to be recruited at intermediates that do not originate from interhomolog recombination events.” Is there good evidence for this? Do COSA-1 foci in synapsis-defective germlines rely on RAD-51? What about the possibility that COSA-1 foci in these synapsis-defective genotypes are associated with interhomolog recombination intermediates that become noncrossovers?

13. Is it the case that irradiation of spo-11 males blocks the accumulation of late pachytene COSA-1 foci associated with SPO-11 removal? I believe this is the case, but the authors may want to bring this out in their discussion of pro-CO markers in late prophase nuclei being caused by “absence of DSBs per se rather than SPO-11 specifically”. The Discussion otherwise is thorough and helpful.

Minor Comments

a. The Introduction has a sentence that begins “COs shuffle maternal and paternal information and provide a physical connection between…” I suggest removing “ shuffle maternal and paternal information and” (mom and dad genetic information shuffles anyhow through independent assortment, and this study is about the more essential and cooler purpose of COs - physical connection, chromosome segregation, genome integrity).

b. The DAPI signal throughout all figures should be changed (more cyan) to match that of Fig S2C.

c. The following sentences made me confused because I could not determine what the X chromosome had to do with interpreting the <5 foci result: “Compared to hermaphrodites [54, 55], a significant cohort of nuclei with <5 COSA-1 foci was observed in males (Fig. 6A). Since the single X chromosome present in males does not undergo CO recombination, this phenotype was somewhat surprising and…” I think it would be much clearer if “Since the single X chromosome present in males does not undergo CO recombination” was removed from the second sentence.

d. “We noticed that under TOP-2 depletion conditions, the average number of COSA-

foci was further enhanced compared to SPO-11 depletion only (Fig. 5A), which we partly

attribute to the possible generation of polyploid cells arising from impaired cell division due to the loss of TOP-2.” This sentence should have a citation I think for the data showing that loss of TOP-2 leads to polyploid nuclei in the germline.

e. Author summary has “by the topoisomerase SPO-11…” the SPO-11 enzyme might not be considered a true topoisomerase by many since it doesn’t carry out the rejoining reaction on broken DNA– to avoid confusing the less specialized reader or irritating the more specialized reader may want to instead write “topoisomerase-like”

f. Author summary could use a little revising to tell the reader that crossovers lead to physical attachments between chromosomes which are important for their proper segregation on the meiosis I spindle.

g. Figure 1 and Figure 2 legend titles refer to crossover numbers but measure COSA-1 foci. I understand that the authors are using COSA-1 foci as a proxy for crossovers, but particularly in this case where the authors have some evidence for COSA-1 foci that do not correspond to a crossover intermediate it is more rigorous to state “does not alter COSA-1 foci numbers”.

Reviewer #2: This manuscript by Engebrecht et al. investigates crossover control during male meiosis in C. elegans. C. elegans is known for its stringent crossover regulation, limiting the number of crossovers to one per homologous chromosome pair. The authors demonstrate that this control is more relaxed in males than in hermaphrodites. They show that in male spermatocytes, the number of COSA-1-makred crossover sites increases linearly with increasing doses of irradiation, in contrast to the sigmoidal response previously reported for hermaphrodites. The study also reveals that COSA-1 foci form in late pachytene even in the absence of SPO-11 and the synaptonemal complex, and that this response is also sexually dimorphic.

The paper is largely observational and does not elucidate the mechanisms underlying the observed sexual dimorphism or its significance. Nevertheless, this manuscript is well written and presents data that are of interest to the field. Specific points are listed below:

1. Figure 3: While the difference in the kinetics of COSA-1 foci appearance following irradiation between males and hermaphrodites is clear, it may not be readily apparent to readers who are unfamiliar with the data presented in Yokoo et al., 2012.

2. Figure 4C: The % viable progeny sired by cosa-1 mutant males is comparable to that by spo-11 mutant males, supporting the conclusion that the improved viability by spo-11 males is not due to conversion of DSBs into crossovers. If this is not the cause, what alternative explanation might account for the increased viability?

3. Figure 4A: Please provide a schematic showing how the regions (zones) are divided.

4. Figure 4B: I wonder whether there is a way to integrate the quantifications for RAD-51 (4A) and COSA-1 (right panel) with the zoning information. Presenting both datasets on the same graph would clarify that COSA-1 foci form independently of RAD-51. Additionally, since RAD-51 and COSA-1 typically do not co-localize in wild type, the lack of co-localization may be expected and worth noting.

5. Page 14, In the final sentence starting with “This suggests that upon reduced physiological DSB levels”, an alternative explanation for the unaffected COSA-1 loading in him-5, him-17, and dsb-2 mutants is that the loss of these genes does not impair DSB formation in males to the same extent as in hermaphrodites. It would be important to demonstrate this by showing RAD-51 staining in these mutants.

6. Page 15, the final sentence of the Results about the possibility of polyploid cells: is there any sign of polyploidy such as enlarged nuclei in the progenitor zone?

7. Page 17, the first sentence: Cahoon et al., (2019 Genetics, Figure 2) reported that pro-CO factors such as MSH-5 and ZHP-3 form foci in spo-11; syp-1 and spo-11; syp-2 mutant hermaphrodites. Thus, the observation of COSA-1 foci in males in the absence of SYP-2 and SPO-11 may not be entirely surprising. Although Cahoon et al. did not quantify these foci, their figure suggests that only a single focus is typically presented. This is very different from the broader distribution of COSA-1 foci shown in Figure 7B of the current manuscript. Could this be due to residual SPO-11 activity despite auxin treatment? Indeed, the authors discuss this possibility on page 21. It would be important to formally test this possibility using a genetic null.

Reviewer #3: This manuscript examines meiotic double strand break formation and crossover in the C. elegans males highlighting important different between the sexes that cannot be explained by sex chromosome composition or speed of meiotic prophase progression. They also show that crossover designation factors accumulate one chromosomes in late meiosis I male nuclei in the absence of spo-11 and indepenendently of topoisomerases, suggesting that there may be male specific chromatin configurations that are recognized by this machinery when normal crossovers are not present (and presumably there is an excess of these proteins). Overall, the studies are interesting and well controlled and should be of strong interest to the field.

I have just a few minor comment for the authors to consider/ address:

1. In the shown examples of 6 COSA-1 foci, it seems that one of the foci is smaller than the others. Is this always the case? Could it be that this is on the X? Given that the Glo:COSA-1 is brighter than the previous transgenes, maybe this is recognizing an intersister exchange. Can staining with anti-HTP-3 show that the extra COSA-1 foci happen to be on a chromosome that already has another COSA-1 focus?

2. The problem with plotting the data as 5, 10, 50, 100Gy is that the dose is not linear, so actually the dose curve is asymptotic. The differences here could be due to eliciting different alternative repair pathways in the male germ line. IR breaks and SPO-11-induced breaks are fundamentally different in their initial processing.

3. p. 12. The statement “no colocalization with OLLAS::COSA-1, suggesting that chromatin recruitment of COSA-1 occurs independently of prior RAD-51 foci formation” is challenging. These two proteins do not colocalize in hermaphrodites either using traditional staining methods. RAD-51 is effectively replaced by MSH-5 and then COSA-1. That said, the following experiments showing lack of COSA-1 in the spo-11::AID rad-51 null do strongly support this conclusion, so perhaps just rephrase this sentence?

4. p. 13 spo-11::AID rad-51 TIR1::mRuby. This genotype is confusing – is TIR1 also on the same chromosome? What promoter is driving the TIR-1 here?

5. It is not just that the COSA-1 foci are independent of SPO-11 but in the spo-=11::AID on auxin, there is an altered distribution of COSA-1 foci. DO the authors have any thoughts or explanation for this? (this should also be mentioned specifically in the text).

6. In figure 5C, please label not just GFP and OLLAS, but the specific protein to which they are attached.

Also note that red-green colors are not color-blind friendly and change accordingly, if possible.

7. The rightmost panel in Figure 5B appears to be mislabeled since there is not DNA stain there and it is unclear which foci these are.

8. Do RMH-1 foci show the same “extra” CO that were shown in Fig 1 for COSA-1?

9. It is unclear whether the proximal CO factor foci in the absence of SPO-11 are appearing AFTER late pachytene or still in LP?

10. ATM-1 in hermaphrodite worms appears not to promote extra breaks as in mice (ref 75), although it is likely part of a feedback mechanism on DSBs.

11. Meneely show that Him-5 reduces the total number of DSBs in hermaphrodites including on autosomes. It is this feature which appears to be active in the males.

12. Ref 10 does suggest that there can be 6 COSA-1 foci in the male germ line (see Fig 5, source data). Please reference this accordingly.

13. “more RAD-51 foci are observed in males compared to hermaphrodite germ cells [10, 22, 23].” Ref 22 shows that number of RAD-51 foci in rad-54 mutants. Those numbers are close to published numbers for hermaphrodites. So I am not sure this is true.

**Have all data underlying the figures and results presented in the manuscript been provided?**

Reviewer #1: Yes

Reviewer #2: Yes

Reviewer #3: Yes

PLOS authors have the option to publish the peer review history of their article (what does this mean? ). If published, this will include your full peer review and any attached files.

**Do you want your identity to be public for this peer review?** For information about this choice, including consent withdrawal, please see our Privacy Policy .

Reviewer #1: No

Reviewer #2: No

Reviewer #3: No

**Figure resubmission:**
---

## [Editor Report · Decision Letter 1]

10 Oct 2025

Dear Dr Engebrecht,

We are pleased to inform you that your manuscript entitled "Loss of meiotic double strand breaks triggers recruitment of recombination-independent pro-crossover factors in C. elegans spermatogenesis" has been editorially accepted for publication in PLOS Genetics. Congratulations!

Yours sincerely,

Monica P. Colaiácovo

Section Editor

PLOS Genetics

Aimée Dudley

Editor-in-Chief

PLOS Genetics

Anne Goriely

Editor-in-Chief

PLOS Genetics

BlueSky: @plos.bsky.social

Comments from the reviewers (if applicable):

**Data Deposition**

http://datadryad.org/submit?journalID=pgenetics&manu=PGENETICS-D-25-00694R1

**Press Queries**

---

## [Editor Report · Acceptance letter]

PGENETICS-D-25-00694R1

Loss of meiotic double strand breaks triggers recruitment of recombination-independent pro-crossover factors in *C. elegans* spermatogenesis

Dear Dr Engebrecht,

We are pleased to inform you that your manuscript entitled "Loss of meiotic double strand breaks triggers recruitment of recombination-independent pro-crossover factors in *C. elegans* spermatogenesis" has been formally accepted for publication in PLOS Genetics! Your manuscript is now with our production department and you will be notified of the publication date in due course.

With kind regards,

Zsofia Freund

PLOS Genetics

On behalf of:
